# Interaction of Proteins Involved in Neuronal Proteinopathies

**DOI:** 10.3390/life13101954

**Published:** 2023-09-23

**Authors:** Konstantin Y. Kulichikhin, Oksana A. Malikova, Anastasia E. Zobnina, Natalia M. Zalutskaya, Aleksandr A. Rubel

**Affiliations:** 1Laboratory of Amyloid Biology, St. Petersburg State University, 199034 St. Petersburg, Russia; oks_malik@mail.ru (O.A.M.); agrad74@mail.ru (A.E.Z.); 2V.M. Bekhterev National Medical Research Center for Psychiatry and Neurology, 192019 St. Petersburg, Russia; nzalutskaya@yandex.ru

**Keywords:** amyloid β peptide, tau protein, α-synuclein, TDP-43, amylin, protein misfolding, proteinopathy, disease-specific conformation, protein interaction

## Abstract

Proteinopathy is characterized by the accumulation of aggregates of a specific protein in a target organ, tissue, or cell. The aggregation of the same protein can cause different pathologies as single protein can adopt various amyloidogenic, disease-specific conformations. The conformation governs the interaction of amyloid aggregates with other proteins that are prone to misfolding and, thus, determines disease-specific spectrum of concomitant pathologies. In this regard, a detailed description of amyloid protein conformation as well as spectrum of its interaction with other proteins become a key point for drafting of precise description of the disease. The majority of clinical cases of neuronal proteinopathies is caused by the aggregation of rather limited range of amyloidogenic proteins. Here, we provided the characterization of pathologies, related to the aggregation of amyloid β peptide, tau protein, α-synuclein, TDP-43, and amylin, giving a short description of pathologies themselves, recent advances in elucidation of misfolded protein conformation, with emphasis on those protein aggregates extracted from biological samples, what is known about the interaction of this proteins, and the influence of this interaction on the progression of underlying disease and comorbidities.

## 1. Introduction

Amyloid protein aggregation is the cause of a number of incurable and lethal diseases commonly names as amyloidoses or proteinopathies. In humans, more than 50 proteins and peptides prone to aggregation and even larger number of pathologies related to protein amyloidization have been discovered [1]. Despite this diversity, in case of neurodegenerative proteinopathies, nearly all of them can be described by the presence of mostly one of four protein aggregates—amyloid β-peptide (Aβ), τ-protein (tau), α-synuclein (α-Syn), and TAR DNA-binding protein 43 (transactive response DNA-binding protein 43 kDa, or TDP-43)—with morphology and distribution specific for each pathology [2]. Extracellular Aβ plaques (senile plaques) and cytoplasmic aggregates of phosphorylated tau (neurofibrillary tangles–NFTs and neuropil threads–NTs) are characteristic features of Alzheimer’s disease (AD) [3,4]. Various forms of tauopathies (Pick’s disease—PiD, corticobasal degeneration—CBD, progressive supranuclear palsy—PSP, argyrophilic grain disease—AGD, primary age-related tauopathy—PART, and neurofibrillary tangle-predominant dementia) commonly named as frontotemporal lobar degeneration with tau pathology (FTLD-tau) are characterized by cytoplasmic aggregates of phosphorylated tau [5,6]. The deposits of aggregated tau are different in morphology and they are localized in neuronal and glial cells in distinct parts of the brain depending on pathology [7]. Similarly, neuronal cytoplasmic α-Syn aggregates in forms of Lewy bodies (LB) and Lewy neurites (LN) are a neuropathological hallmark of Lewy body diseases (LBDs) that covers Parkinson’s disease (PD), Parkinson’s disease dementia (PDD), and dementia with Lewy bodies (DLB). PD and DLB differs by the location of where α-Syn aggregation begins and direction where the pathology spread [8]. Multiply system atrophy (MSA) is characterized by filamentous α-Syn inclusions in olygodendrocytes with smaller amount of aggregated α-Syn also presented in neuronal cells [9]. Intracellular aggregates of phosphorylated TDP-43 in nuclei and cytoplasm of neurons and glial cells cause FTLD-TDP [10] and certain forms of amyotrophic lateral sclerosis (ALS) [11]. Taking into account similarities in genetic background, pathological hallmark, and clinical features of FTLD and ALS, they are frequently considered to be part of a common ALS-FTLD disease continuum [12].

As mentioned above, aggregation of the same protein can cause different pathologies. This phenomenon can be explained by the ability of amyloidogenic proteins to adopt various amyloid, disease-specific conformations. This conformation governs the interaction of amyloid aggregates with other proteins that are prone to aggregation and thus can determine disease-specific spectrum of concomitant pathologies. Co-pathologies can significantly increase the severity of underlying disease and establishing the range of possible co-pathologies is of extreme importance for setting up the correct treatment plan. In this regard, the detailed description of amyloid protein conformation as well as spectrum of its interaction with other proteins that are prone to aggregation become a key point for drafting of precise description of any protein misfolding disease.

In this review, we focused on what is known to date about the interaction between Aβ, tau, α-Syn, TDP-43, and amylin; recent advances in elucidation of atomic structure of amyloid assemblies formed by these proteins; and we discussed how this information can help in understanding of disease nature and developing new approaches for disease diagnostics and treatment. We also implemented into discussion peptide hormone amylin, also known as islet amyloid polypeptide (IAPP). This hormone is produced in pancreatic β-cells, and its amyloid aggregationin pancreas is a pathologic hallmark of type II diabetes mellitus (T2DM)—the most widely spread form of diabetes worldwide [13,14]. Despite amylin aggregation taking place out of nervous system, T2DM is proved to be a risk factor for various neuronal proteopathies, in particular, AD [15,16,17,18] and PD [19,20].

## 2. Pathologies Related to the Aggregation of Proteins

### 2.1. Tau Protein-Related Pathologies

Tau is mainly a neuronal microtube-associated protein encoded by the *MAPT* gene. This gene has 16 exons. Alternative splicing of exon 10 results in tau isoforms with three or four microtube-bindings repeats named 3R and 4R isoforms. Alternative splicing of exons 2 and 3 leads to isoforms with no (0N), one (1N), or two (2N) N-terminal inserts that in combination with 3R and 4R varieties gives rise to 6 isoforms 0N3R, 1N3R, 2N3R, 0N4R, 1N4R, and 2N4R (see Figure 1 and [21]).

In tauopathies, tau undergoes post-translational modifications (PTMs) leading to the destabilization of its interaction with microtubules and appearance of self-aggregation capacity. For example, pathological tau extracted from AD brain represented 95 PTMs (55 phosphorylation sites, 17 ubiquitination sites, 19 acetylation cites, and 4 methylation sites) at 88 different amino acid residues [22]. Tau pathologies are diverse in terms of morphology of inclusions and their cellular localization [21] and can be distinguished by different variants of silver-staining techniques [23]. This suggests that a particular pathology may be associated with the aggregation of specific tau isoforms as well as with the specific profile of tau PTMs. Indeed, AD and PART are characterized by inclusions formed by both 3R and 4R tau isoforms, aggregates originated from PiD brain are formed exclusively by 3R tau, whereas in case of PSP, CBD, and AGD, the pathology is determined by misfolding of 4R isoforms [21].

The antibody 12E8 that detects pSer262/pSer356 tau has been found to be applicable for discrimination between PiD and AD or CBD. This epitope is present in AD and CBD, but absent in PiD [24,25], representing an example of disease-specific PTM. The identification of these specific PTMs may be important for the development of molecular tools for diagnosis of tauopathies [26]. The combination of tau isoforms and PTMs that are unique for each pathology explains why tau adopts disease-specific molecular conformations [27]. Further evidence on disease-specific strains of pathological tau were obtained in mice models. It has been shown that intracerebral injection of human brain extracts containing various pathological tau (AGD, CBD, PSP, PiD, AD) into the brain of transgenic (AGD, CBD, PiD [28]; AD, CBD [29]) and non-transgenic mice (AGD, CBD, PiD [28]; AD, CBD, PSP [30]) led to the development of cell-type specific and morphologically distinct tau pathologies that phenocopy the corresponding tauopathies, while the injection of synthetic tau preformed fibrils did not seed endogenous tau aggregates [31].

Recently, cryo-EM was applied to characterize the strains of misfolded tau specific for AD [32] and PiD [25]. AD-derived tau fibrils contained both 3R and 4R tau and were made of two identical C-shaped protofilaments with amyloid core formed by eight β-strains in the region of 306–378 a.a. residues (see Figure 2 and [32]).

The fibrils were presented by two structural polymorphs–symmetric paired helical and asymmetric straight (see Figure 3 and [32]). The polymorphs were different in their inter-protofilament interface. In paired helical filaments, it was formed by anti-parallel stacking of a.a. residues 332–336, whereas in straight polymorph, the backbones of two protofilaments were closest to each other between the region of 321–324 a.a. of the first and 313–317 a.a. of the second protofilament (Figure 3). This interface of straight polymorph is additionally stabilized by additional interactions that involved side chains of residues K317, T319, K321, and probably N-terminal region of residues 7–9 [32]. The authors proposed this interaction to form the epitope of AD-specific tau antibodies Alz-50 and MC-1 that is formed by two discontinuous regions 7–9 and 313–322 residues [32].

Structures of filaments from PiD revealed completely different tau protein fold compared with described for AD [25]. PiD-derived tau fibrils were composed exclusively by 3R tau and were represented by two polymorphs: narrow (93%) and wide (7%) [25]. PiD’s protofilaments had J-shaped morphology that was distinctly different from AD-derived C-shaped protofilaments. Narrow polymorph was formed by single protofilament (Figure 3), whereas wide polymorph was formed by the association of two narrow fibrils at their distal tip (see Extended Data Figure 3 in [25]). Amyloid core is formed by nine β-strains and consists of residues 254–378 of 3R tau (based on numbering of 2N4R human tau isoform) (Figure 2). Inside this amyloid core, 4R tau isoforms contain the second microtube-binding repeat (275–306 a.a. residues), that is absent in 3R tau. This additional region prevents 4R isoforms from adopting PiD-specific folding, thus giving the explanation why PiD is characterized by aggregation of 3R tau only. Indeed, PiD-derived tau filaments can effectively seed in vitro recombinant 3R but not 4R tau [25]. Conformational differences between PiD- and AD-derived tau assemblies also explain the differences in Ser262 phosphorylation between these diseases, mentioned above: being involved into amyloid core, this residue becomes protected from the phosphorylation in PiD, whereas in AD, it remains accessible for PTM [25]. Heparin-induced tau fibrils were polymorphic and structurally distinct from PiD and AD-derived fibrils [33], which may also explain why their injection into mice brain did not seed endogenous tau aggregation [31]. Thus, cryo-EM analysis of tau fibrils from in vitro reaction and from patients with AD and PiD confirmed the hypothesis that different conformers (strains) of aggregated tau give rise to the distinct clinical phenotypes of different tauopathies, as have been previously proposed for prion strains [34]. Recently, conformation-selective tau antibodies GT-7 and GT-38 were developed [35]. These antibodies demonstrated selective binding to AD-derived tau aggregates and no immunoreactivity to aggregates in CBD, PSP, and PiD, providing an additional support to the concept of disease-specific tau conformations [35].

Tau pathology can manifest at early age and is widely spread in population. In a study, where 2332 nonselected brains from 1- to 100-year-old individuals were examined, only 10 cases showed complete absence of abnormally phosphorylated tau [36]. These 10 cases were mostly corresponded to the individuals under the age of 10 [36] and the earliest non-fibrillar abnormal phosphorylated tau protein accumulation called pretanlges was identified in the brain of a 6-year-old boy [37]. The first NFT was detectable at age between 10 and 20 years; at 40 years of age, most individuals already possessed tau pathology, whereas at an age of over 91 years, nearly all people showed at least Braak-NFT stage I. This early onset and almost uniform distribution of tau pathology among human population suggest that tau aggregation may be the part of the natural aging process [36].

### 2.2. Amyloid-β-Related Pathologies

Amyloid-β (Aβ) peptides are the products of sequential proteolysis of the amyloid precursor protein (APP) by β- and γ-secretases (Figure 4). Taking into account multiply cleavage cites for both β- and γ-secretases, proteolysis of APP generate Aβ peptides of varying lengths with Aβ40 and Aβ42 as the most abundant variants [38]. These peptides, especially Aβ42, are highly propensive to aggregation. Extracellular aggregation of Aβ in brain parenchyma leads to formation amyloid plaques. Together with NFT formed by hyperphosphorylated tau, they form constitute neuritic plaque–distinct pathological hallmark of AD [3,4]. Aβ also forms deposits at cerebral vasculature known as cerebral amyloid angiopathy (CAA) [39]. Parenchymal Aβ deposits are predominantly formed by Aβ42 and Aβ43 peptides whereas in vascular deposits shorter variants from Aβ36 to Aβ41 are present [39].

The function of Aβ in the brain is not yet clear. Nevertheless, recently, it has been shown that Aβ can take a part in regulation of synapse numbers and synaptic transmission in cultured human neurons [40]. The accumulation of Aβ in the brain is prevented by coordinate operation of several cleaning systems that include (1) transport through blood–brain barrier (BBB), (2) cellular uptake and subsequent proteolysis, (3) clearance through CSF, and (4) glymphatic pathway [41].

Factors affecting normal cleaning of Aβ from the brain or leading to the increase in Aβ production as well as to the shift of Aβ42/Aβ40 ratio can induce Aβ aggregation in the brain. For example, mutations in genes *PSEN1* and *PSEN2* that encoding subunit 1 and subunit 2 of γ-secretase, correspondingly, lead to elevated production of Aβ42 and thus increase the risk of AD.

On the other hand, mutation in APP has more specific impact on the pathology. The substitution of a single amino acid in APP can change (1) the pattern of its proteolysis and, consequently, the spectrum of proteolytic fragments and their ratio, (2) the aggregation propensity of Aβ, as well as (3) its affinity to receptors and accessibility to proteases involved in the clearance of Aβ in the brain. Several APP mutations were shown to lead to CAA with little or no Aβ accumulation in brain parenchyma (E693K–Italian, E693Q–Dutch, D694N–Iowa, L705V–Piedmont. For detailed descriptions of these mutations and mutations mentioned below, see [42] and references therein). Mutation E693K leads to decrease of Aβ42/Aβ40 ratio, whereas mutation E693Q results in elevated aggregation propensity of Aβ40. APP mutations that lead to increase of overall Aβ production such as KM670/671NL (Swedish) lead to elevation of both AD and CAA pathology, whereas mutations that shift Aβ42/Aβ40 ratio toward Aβ42 (for example, T714I–Austrian, V717I–London, etc.) trigger the pathological process toward AD. Aβ40 peptide derived from APP possessing the mutation E693G (Arctic) is characterized by increased propensity to aggregation and, at the same time, confers increased resistance of peptide to proteolysis. Additional genetic factor affecting Aβ pathology is *ApoE* genotype. *ApoE4* allele is risk factor for AD, while *ApoE2* is believed to be AD-protective. On the other hand, *ApoE2* is risk factor for CAA [43]. Nevertheless, it is necessary to point out that hereditary forms represent only a small fraction of the total number of AD cases, while usually pathologies due to Aβ aggregation do occur spontaneously.

Similarly with tau, misfolding of Aβ results in appearance of various amyloid strains. In vitro aggregated Aβ40 and Aβ42 formed morphologically distinct fibrils that being injected intracerebrally into mouse brain formed there morphologically distinct pathology [44]. The inoculation of transgenic mice overproducing human Aβ with brain extract from AD cases with Arctic or Swedish APP mutation exhibited the development of pathology specific for each mutation and this divergent pattern of pathology was reproduced during the second passage of aggregated Aβ in mice [45]. Strain-specific features were determined by conformation of aggregated Aβ injected into mouse brain rather than the APP genotype of recipient mice [46]. The investigation of in vivo-formed amyloids derived from different patients with AD using conformation-specific luminescent dyes revealed marked variation of spectral attributes between samples from patients with sporadic or familiar AD, but they were similar in different cortical regions of individual patient [47].

The application of cryo-EM allowed to discover diverse structures of in vitro-formed fibrils of Aβ42 that differed in chirality as well as in the shape of the protofilaments [48,49,50]; one of these structures is presented on Figure 5. Observed diversity in folding pattern may reflect the difference in aggregation condition in vitro such as aggregation buffer pH and/or composition. Additionally, the analysis of Aβ fibrils from human brain [51,52] revealed significant differences between assemblies derived from in vitro reaction and from those extracted ex vivo from biological samples (Figure 5). Significant differences were also found between fibrils derived from sporadic AD, familiar AD, and CAA [51,52]. Moreover, several aggregated Aβ polymorphs may be characterized in the same biological sample [51], as was also shown for in vivo aggregated tau protein [25,32].

Brain parenchyma-derived fibrils from AD were made of two identical S-shaped protofilaments, were left-hand twisted, and Aβ42 was the most predominant Aβ species [51].

Two types of fibrils were described and these types differed in number and location of β-strands along peptide chain: type I (5 β-strands) was predominant in sporadic AD, whereas type II (4 β-strands) fibrils were characteristic feature of familiar AD as well as of Aβ pathologies accompanying other neurodegenerative diseases [51]. It is necessary to mention that familiar AD cases used in the research mentioned above were characterized by mutation in *PSEN1* or characterized by mutation V717F in APP; both of them did not affect amino acid sequence of Aβ [51]. On the other hand, in vascular deposits-derived fibrils, the peptide fold was C-shaped, protofilaments had a right-hand twist, and were predominantly formed by shorter forms Aβ36–Aβ40 with minimal amount of Aβ42 [52]. Four β-strands along the peptide chain were identified and their localization differs from those previously described for AD [52]. CAA-derived fibrils can be comprised by one, two, or three protofibrils representing three different types of fibril architecture–type I, II, or III, correspondingly [52]. Thus, a series of experiments confirm the difference between in vitro and in vivo Aβ assemblies as well as the heterogeneity of pathogenic Aβ conformations among different AD variants and CAA.

Aβ aggregation can be firstly observed at age of 11–20 years and further the frequency and distribution of plaques increase with age and approx. 80% of all autopsy cases showed Aβ-deposits at age of 80 [53]. Nevertheless, 20% of people up to the age 100 years do not develop amyloid plaques at all [53]. Aβ plagues first appeared in neocortical area and then expand into further brain regions in a distinct hierarchical sequence [54].

### 2.3. α-Synuclein-Related Pathologies

The α-synuclein (α-Syn) is a protein composed of 140 amino acid residues with a molecular weight of approximately 15 kDa and encoded by the *SNCA* gene (see Figure 6 and [55,56]). It is present in various compartments of neuronal cells such as synaptic terminals, nuclei, mitochondria, endoplasmic reticulum, Golgi apparatus but its exact function is not completely clear [57]. Residues 7–87 of α-Syn represents the lipid-binding domain consisted of seven imperfect repeats with the consensus sequence KTKEGV. These repeats partially overlap with central hydrophobic region (residues 61–95) known as non-amyloid component (NAC) that determines aggregation capacity of α-Syn. C-terminal region (residues 96–140) is rich in acidic residues and negatively charged, and its cleavage leads to enhanced propensity of α-Syn to form fibrils [58]. Amyloid core of α-Syn filaments includes the region of amino acid residues 30–110 [59,60]. This includes five of seven tandem repeats, NAC region as well as a part of the carboxyl-terminal acidic region. The aggregation of α-Syn results in impaired axonal transport, mitochondrial function, and calcium homeostasis, as well as in increased oxidative stress and neuroinflammation [61].

In Lewy bodies disease (LBD), which encompasses Parkinson’s disease (PD), Parkinson’s disease dementia (PDD), and dementia with Lewy bodies (DLB), α-Syn inclusions are localized in nerve cells of central and peripheral nervous system as Lewy bodies (LB) and Lewy neurites (LN) [8,9], whereas filamentous α-Syn aggregates in oligodendrocytes, also known as glial cytoplasmic inclusions (GCI) or Papp-Lantos bodies, are distinct pathological hallmark of MSA [9,62]. Neuronal inclusion can also be present in MSA, but they are rather rare.

Recombinant α-Syn aggregated in vitro can form oligomeric and fibrillar assemblies [63,64], and these assemblies can adopt various conformations [65,66]. Two morphologically, biochemically, and functionally distinct polymorphs of in vitro-aggregated α-Syn were characterized and named fibrils or rod polymorph and ribbons or twister polymorph based on their specific morphological features [66,67]. Rod polymorph formed cylindrical, non-twisted [67], or apparently non-twisted assemblies [66], whereas twister polymorph’s assemblies were flat and twisted. In addition, these polymorphs differed in the spectrum of proteolytic fragments after the digestion by proteinase K, in seeding and propagation propensity in vitro and in vivo, as well as in their toxicity for neuroblastoma cell culture [67]. The injection of fibrils polymorph assemblies into the brain of rats overexpressing human α-Syn resulted in progressive motor impairment and progressive cell death, whereas the application of ribbons caused the development of pathology similar to PD and MSA [68]. Based on these observations, different strains of pathologic α-Syn were proposed to be responsible for heterogeneous nature of synucleinopathies.

Indeed, pathological α-Syn extracted from neuronal LB and GCI (associated with DLB and MSA, correspondingly) were conformationally and biologically distinct differing in tolerance to proteinase K, in recognition by α-Syn-strain-specific monoclonal antibodies, in propensity to seed α-Syn in vitro and in vivo, and in capacity to induce α-Syn-pathology [69]. The GCI-derived strain had approximately 1000-fold higher seeding activity and aggregation of α-Syn in oligodendrocytes rather than in neurons resulted in conversion of misfolded α-Syn into aggressive MSA-specific strain [69]. Thus, distinct pathological conformations of α-Syn can be determined by different unique cellular milieu.

Van der Perren and co-workers [70] have isolated pathological α-Syn strains from patients with PD, DLB, and MSA. It was shown that MSA and PD strains formed aggregates of similar morphology with ribbons (twister) polymorph. In addition, these strains showed similar pattern of proteolysis by proteinase K and in cell culture—similar distribution of phosphorylated α-Syn deposits. In a mouse model, MSA strains were much more effective than PD-derived aggregates in spreading and inducing α-Syn pathology, inflammation, neurodegeneration, and motor deficit. DLB strains of α-Syn formed non-twisted assemblies [70] that were similar to fibrils (rod) polymorph. They were less pathogenic in neuronal cells and in a mouse model, displaying no or very modest neuropathological features [70]. DLB-derived α-Syn aggregates were both sarkosyl- and SDS-insoluble whereas assembles from patients with MSA were resistant to sarkosyl only [71,72].

Shahnawaz and co-workers [73] discovered the possibility of protein misfolding cyclic amplification technique (PMCA) to discriminate between samples of brain tissue or cerebrospinal fluid (CSF) from patients with PD and samples from patients with MSA by differences in maximum fluorescence values reached in PMCA reaction and by differential binding of luminescent conjugated oligothiophenes (LCOs). Additionally, proteinase K-digested PD- and MSA-derived fibrils differed in epitope mapping by antibodies specific for distinct regions of α-Syn [73].

Cryo-EM analysis of in vitro assembled rod (fibrils) and twister (ribbons) α-Syn polymorphs [66] revealed that these assemblies consisted of two identical protofilaments. The polymorphs share common protofilament kernel structure–a bent β-arch formed by the region 57–67 a.a. residues, but differ in their inter-protofilament interface (Figure 7).

In the rod (fibril) polymorph, protofilament interaction is provided by preNAC core region (47–56 a.a. residues), whereas in the twister (ribbon) polymorph, a pair of protofilaments pack around NAC core region (68–78 a.a. residues) [66]. The preNAC region is associated with several PD familiar mutations (E46K, H50Q, G51D, A53E, A53T, A53V) and implementation of any of these mutations would probably disrupt interaction between protofibrils and prevent misfolding to rod polymorph, thus shifting to the formation of twister or other PD-specific polymorphs [66,74].

In contrast to in vitro α-Syn assemblies, the inclusions from the brains of patients with MSA were made of two types of twisted filaments, each of them consisted of two different protofilaments thus representing four distinct variants of conformations (see Figure 8 and [72]). Non-proteinaceous molecules of unknown chemical nature were present at the interface of the two protofilaments in both types of filaments. Additionally, N-terminal region (14–37 a.a. residues) is mostly implemented into protofilament core of MSA-derived fibrils that is rather unusual for in vitro aggregated α-Syn [72]. On the other hand, α-Syn filaments from the brains of individuals with PD, PDD, and DLB were mostly not twisted, composed of single protofilament and their folding pattern (Lewy fold) was completely different from both in vitro aggregated α-Syn and ex vivo extracted MSA-derived assemblies (see Figure 8 and [75]).

Thus, disease-specific molecular conformers of assembled α-Syn have been described [72,75]. Similarly with tau and Aβ, the nature of exact synucleinopathy is determined by specific strain of pathological α-syn. On the other hand, distinct pathological conformation is determined by specific aggregation condition in vitro or unique cellular milieu in vivo.

### 2.4. TDP-43-Related Pathologies

TDP-43 is a 43 kDa heterogenous nuclear ribonuclear protein (hnRNP) consists of 414 a.a. residues and is encoded by the *TARDBP* gene. TDP-43 is synthesized in cytoplasm and then shuttled into the nucleus where it primarily resides to perform its physiological function [76,77]. TPD-43 is composed of N-terminus, nuclear localization signal (NLS), two centrally located RNA-binding domains (RBD1 and RBD2), and a C-terminal glycine-rich domain (GRD) that contains low-complexity region prone to amyloid aggregation. RBD2 contains nuclear export signal (NES) that together with NLS provides TDP-43 shuttling between nucleus and cytoplasm (see Figure 9 and [76,77]).

The function of TDP-43 is regulation of gene expression and RNA processing including RNA splicing, mRNA turnover, RNA trafficking, and microRNA biogenesis [78]. In condition of cellular stress, TDP-43 is located in stress granules being associated with ribosomes and participate in blockade of translation and promotion of cytoprotective proteins synthesis [79,80]. Importantly, the functional form of TDP-43 in vivo is oligomeric, assembled in helical filaments [81]. Physiological oligomerization is mediated by the interaction of N- and C-domains in head-to-tail fashion. Such an N-terminal-driven oligomerization spatially separates the adjoining aggregation-prone C-terminal GRD of neighboring TDP-43 monomers, thereby preventing GRD intermolecular interaction and thus antagonizing the formation of pathological aggregates [81].

Under normal physiological condition, the oligomeric and monomeric TDP-43 are in equilibrium. The disturbance of this equilibrium in the cytoplasm may result in proteolytic cleavage of monomeric TDP-43 and the generation of C-terminal fragments which are highly prone to aggregation due to low-complexity GRD and are able to initiate formation of irreversible pathologic inclusions [81].

Compared with other pathogenic proteins, the amyloidogenic nature of TDP-43 was discovered rather recently. In 2006, Neumann and co-workers [82] identified TDP-43 as the main component of ubiquitin-positive and tau-negative inclusions in brain of patients with FTLD and ALS. PTMs, namely phosphorylation, ubiquitination, and truncation, led to conformational changes of TDP-43. This results in the decreased shuttling of protein to the nucleus, mislocalization into cytoplasm, and promotes aggregation. Cytoplasmic aggregates of TPD-43 were found to be cytotoxic both in vitro and in vivo [83,84] and loss of nuclear TDP-43 modifies chromatin accessibility that results in alteration of gene expression. Additionally, the loss of nuclear TDP-43 promotes neurodegeneration [85,86]. Thus, the aggregation of TDP-43 results in a double detrimental effect for the nerve cell: the loss of normal function of this protein accompanied with toxicity of TPD-43 assemblies.

The recombinant full-length TDP-43 can form structurally stable oligomers that share common epitopes with anti-amyloid oligomer-specific antibody A11 [83]. Such oligomers were present in transgenic TDP-43 mice as well as in brain tissue samples of patients with AD, ALS, and FLTD-TDP [84,87]. Fibrillar assemblies of TDP-43-GRD or full-length TDP-43 were described and characterized [88,89,90].

TDP-43 inclusion bodies in neurons, in neuronal cell processes, and in glial cell are distinctive pathological hallmarks of ALS and FTLD. Based on morphology, cell types, and subcellular distribution, FTLD is classified into four main subtypes, which are described in details elsewhere [10,91,92]. TPD-43 pathology disseminates in sequential pattern that differ between FTLD and ALS and ranged from stage I to stage IV depending on brain regions covered by pathological process, where stage IV represents the most extensive spreading and the most severe pathology [93,94].

Recently, a more complex nature of TDP-43-associated pathologies was discovered. Despite TDP-43 being proposed to be the main component of protein inclusions in ALS, FTLD-TDP, the fibrillar form of TDP-43 derived from the brains was characterized for ALS patients only [90]. In the case of FTLD-TDP, the fibrils were composed of another protein, namely transmembrane protein 106B, or TMEM106B [95]. Mutations in gene encoding TMEM106B were previously recognized as a risk factor for ALS [96] and FTLD [97]. The aggregated form of phosphorylated TDP-43 was present together with TMEM106B in sarkosyl-insoluble fraction of brain extracts from FTLD-TDP donors, but these aggregates were amorphous rather than amyloid-like [95]. It is uncertain whether the presence of TDP-43 fibrils in ALS with FTDP and FTLD-TDP cases stems from the different neuropathological condition or from different extraction procedures. It is also unclear whether TDP-43 aggregates and TMEM106B fibrils co-localize in the patients’ brains, and whether a cross-seeding between TDP-43 and TMEM106B is involved in FTLD-TDP pathogenesis. This finding may refocus pathogenic studies of FTLD-TDP and perhaps other neurodegenerative diseases to include TMEM106B [95].

The progression of TDP-43 pathology may be associated with the mutations in *TARDBP* gene and these mutations are associated predominantly with GRD [77]. Most of these mutations increase the risk of ALS, whereas only few of them are associated with FTLD. This fact is in accordance with the observation that aggregation of TDP-43 may not be the main pathological event in the development of FTLD-TDP pathology [95]. To elucidate the structure of TDP-43 amyloid, in vitro formed aggregates of short GRD fragments [88] or complete GRD [89] and ex vivo-extracted filaments from the brain of ALS with FTLD patients [90] were analyzed by cryo-EM. Patient-derived fibrils were obtained from two independent individuals and from two different brain regions (frontal and motor cortices) and were identical between patients as well as between brain regions [90]. Nevertheless, ex vivo fibrils showed no similarity to assemblies formed in vitro [88,89] in terms of both morphology and folding pattern (Figure 10).

In vitro-formed aggregates demonstrated left-handed twist [88,89]. Assemblies of GRD were represented by single-protofilament fibrils but the core covered the region of 139 a.a. residues (276–414) and formed by fourteen β-strands (see Figure 10 and [89]). This is the largest core among all amyloid fibrils structurally characterized to date [98]. The ex vivo fibrils had a right-handed twist and were also composed of a single protofilament. The ordered filament core spans a.a. residues 282–360 of GRD and adopt double-spiral-shaped fold. The core was formed by ten short β-strands and eight of them were 2–3 a.a. residues long (see Figure 10 and [90]). The surface of ex vivo TPD-43 amyloid fibrils contained shallow and more polar grooves, compared with other amyloid fibrils described. As these surface properties are important for the binding of amyloid imaging ligands, this may explain lower affinity of TPD-43 fibrils to thioflavin-S and ^18^F-flortaucipil that significantly limit visualization of pathogenic TDP-43 and diagnosis of ALS and FTLD [90].

There are 24 reported mutations within the region 282–360 a.a. residues that cause ALS and 18 of them were fully compatible with double spiral fold of ex vivo fibrils [90]. On the other hand, 17 of them were also compatible with the core structure described for in vitro-assembled fibrils of GRD [89], taking into account different compatibility of each specific mutations for each of the proposed models. The difference in folding between in vitro-assembled and brain-extracted filaments may also be determined by PTMs, in particular phosphorylation at Ser409/Ser410 that are present in ex vivo fibrils, but absent in recombinant protein [90]. The question of FTLD-specific strains of TDP-43 remains opened due to discovery of fibrils formed by TMEM106B rather than by TDP-43 in FTLD-TDP brain [95]. On the other hand, Neumann and co-workers [99] showed a striking difference among FTLD-TDP subtypes in the immunoreactivity profile of TDP-43 inclusions with the antiserum specific to pTDP-43 phosphorylated at Ser375. These data suggest a subtype-specific, conformation-dependent binding of pTDP-43 antiserum to TDP-43 aggregates that is consistent with the idea of distinct structural TDP-43 conformers [99].

To sum up, the recent advances in structural elucidation of TDP-43 fibrils significantly contribute to our knowledge about ALS as primary TDP-43-related pathologies and open a perspective in development of accurate disease model, diagnostic tools, and therapeutic approaches. The elucidation of TMEM106B fibrils in FTLD-TDP-derived samples recalls for deeper analysis of the role of TDP-43 in this pathology and TMEM106B in other neurodegenerative diseases.

### 2.5. Amylin

#### 2.5.1. Amylin, T2DM and Neurodegenerative Proteinopathies

Islet amyloid polypeptide (IAPP) or amylin is a peptide hormone that contains 37 a.a. residues. It is released by pancreatic β-cells together with insulin in response to food intake [100]. Amylin acts as satiation signal by reducing gastric acid secretion, slowing down the rate of gastric emptying and reducing pancreatic glucagon secretion [101].

Amylin is initially expressed as pre-pro-amylin (89 a.a. residues) that undergoes proteolytic processing to pro-amylin. This peptide goes through additional cleavage with simultaneous modifications, namely carboxyl amidation of C-terminal tyrosine and disulfide bridge formation between second and seventh cysteine residues, resulting in mature amylin (see Figure 11 and [102,103]).

Amylin, pro-amylin, and its intermediates are prone to aggregation. Under a pathological condition such as T2DM, amylin can form insoluble amyloid plaques in pancreatic islets [104]. Despite T2DM targeting the peripheral organ, it also may facilitate degenerative processes in central nervous system (CNS). Deposits of amylin amyloid were found in the temporal lobe gray matter as well as in blood vessels and perivascular spaces in the brain of T2DM patients [105]. Amylin aggregation in CNS is a result of hyperamylinemia–a pathological hypersecretion of amylin by pancreatic β-cells that is frequently observed at early stages of T2DM [106,107]. Additionally, T2DM patients were characterized by higher level of total tau and phosphorylated tau in CSF as well as by lower bilateral and parietal cortical thickness compared with individuals without T2DM [108]. These pathological hallmarks are non-specific and may be related to various neurodegenerative diseases. Indeed, as mentioned above, T2DM has been identified as risk factor for both AD [15,16,17,18] and PD [19,20]. T2DM, AD, and PD share common pathological features such as insulin resistance, neuroinflammation, and oxidative stress [107,109].

Insulin resistance is characterized by hyperinsulinemia and decreased expression of insulin-degrading enzyme (IDE)–Zn-metalloprotease, which is responsible for cleavage of insulin, amylin, and Aβ [110]. Under T2DM, IDE-mediated degradation of Aβ in the brain may be restricted by decreased IDE activity as well as by elevated concentration of insulin, thus leading to an increase in Aβ concentration, which facilitates its aggregation [111]. Another mechanism establishing the link between T2DM and AD had been proposed by Qui and Zhu [112]. Amylin and Aβ can concurrently bind the complex of calcitonin receptor (CTR) and receptor activity modifying protein 3 (RAMP3). During the late phase of T2DM, when amylin secretion is significantly decreased (hypoamylinemia), CTR-RAMP3 complex bind predominantly Aβ. Binding to CTR-RAMP3 facilitate Aβ oligomerization, thus promoting the development of Aβ pathology [112]. The application of non-aggregated amylin or its clinical analog may prevent the binding of Aβ to CTR-RAMP3, thereby reducing the potency of Aβ to oligomerize [112].

T2DM may predispose to PD-like pathology and can aggravate PD pathology, when present in patients with PD [113]. The presence of T2DM in patients with PD resulted in faster motor progression and cognitive decline, higher striatal dopamine transporter deficit, and higher tau CSF level compared with PD patients without T2DM [113]. Patients with T2DM but without PD were characterized by higher tau and α-Syn CSF level and decreased dopamine transported binding compared to healthy control [113]. Studies on rodent models have displayed that insulin resistance may result in a reduction in surface dopamine transporters [114] and reduced insulin-dependent dopamine release in the striatum [115], giving an explanation of higher motor progression and dopamine-transporter deficit found in T2DM patients. An elevated level of α-Syn may activate glycogen synthase kinase (GSK-3β) [116], which is involved in tau hyperphosphorylation in neurodegenerative pathologies, particularly in AD [117], representing an additional potential link between T2DM and neuronal proteinopathies.

Thus, T2DM may be a significant risk factor for development of neurodegenerative pathologies by the alteration of brain metabolism and protein turnover, activation/inactivation of signaling pathways that may result in elevating level of peptides and proteins prone to aggregation, such as Aβ and α-Syn, or their PTM, such as phosphorylation of tau, dramatically affecting their aggregation capacity. Apart from these, the aggregation of amyloidogenic proteins may be directly seeded by amylin assemblies those presence in the brain of T2DM patients was experimentally confirmed [105]. This issue will be discussed in details below, and for this purpose, what is known to date about the structure of amylin amyloid aggregates should be reviewed.

#### 2.5.2. Structural Characterization of Amylin Amyloid Assemblies

Despite the amino acid sequence of amylin being strongly conserved among mammalian species, amylin amyloidosis can be discovered in primates including humans and in cats, but not in rodents and dogs [102,107,118]. The amyloidogenic properties of amylin was shown to be connected with the region 20–29 a.a. residues [119,120]. Five of six differences between rat and human amylin corresponds to this region. Rat polypeptide has three proline residues there relative to human amylin, which eliminates the aggregation capacity of rodents-type amylin.

The importance of 20–29 a.a. region for development of T2DM has been confirmed by discovering a mutation in the amylin gene that leads to substitution of serine to glycine at position 20 (S20G) [121]. The majority of patients possessing this mutation developed T2DM at younger age than patients from the control group [121]. S20G-amylin exhibited increased aggregation capacity in vitro compared with the wild-type protein [122,123]. The analysis of amylin from 37 wildlife species, which covered 22 varieties of polypeptide, recognized additional regions critical for amylin aggregation [118]. The substitutions N14K, N14D, and H18R were shown to be potent in impeding the aggregation [118]. Interestingly, H18R mutation can completely diminish pro-amyloidogenic effect of S20G substitution. The combination R18/G20 has been described in ursine amylin [124]. The human amylin double mutant H18R, S20G behaved similarly with H18R mutant, i.e., it possessed even less aggregation capacity than wild-type protein [125].

In vitro aggregation of wild type human amylin results in formation of fibrils with different morphology that are presented in reaction mixture at different abundances [126,127]. Wild type human amylin aggregates were presented by twisted fibrils contained of two identical, symmetrically oriented intertwined protofilaments, each built from S-shaped subunits (see Figure 12 and [128,129]). Three cross-β layers were observed at residues 14–20, 26–32 and 35–37 and protofibril interaction interface was formed by the segment of 21–27 a.a. residues together with amidated terminus of Y37 [128,129].

Interestingly, despite very close morphological and folding similarities of amylin fibrils described by Gallardo and co-workers [128] and Röder and co-workers [129], the former researchers obtained left-handed fibrils, whereas the latter characterized the fibrils with right-handed chirality. The modification of amino acid chain led conformational changes in described structure. Recombinant amylin without Y37 amidation and with attached SUMO-tag at N-terminus demonstrated completely different monomer folding with interaction interface shifted to the region 23–27 and 32–33 a.a. residues [130].

The implementation of S20G mutation to wild-type amylin led to even more dramatic structural changes. S20G variant formed two major polymorphs–2PF and 3PF–consisted of two or three interacting protofilaments, correspondingly. Both polymorphs shared a common two-protofilament symmetric core that was distinct from S-shaped wild type structure and the third protofilament of 3PF polymorph showed different from two others protofilaments pattern of monomer folding.

Each core monomer contained four β-strands formed by a.a. residues 15–18, 21–23, 28–31, and 35–36. Most of them partially overlapped β-strands observed in wild-type amylin protofibrils, but the strand formed by a.a. 21–23 was exclusive for S20G mutant. Core protofibril interaction interface is rather short and covered the region 25–27 a.a. residues. The third protofilament partially overlapped the structure observed for wild-type peptide and contained only two β-strands (15–18 and 25–29 a.a. residues) [128]. Thus, the same amylin sequence was able to adopt multiply conformations within the same molecular assembly similarly with ex vivo MSA-specific folds of α-Syn, mentioned above (Figure 8 and [72]). It is proposed that the presence of two distinct backbone conformations may be the reason of increased aggregation propensity of S20G amylin [128]. Greater fibril stability of S20G variant relative to wild-type protein was also predicted for non-amidated at Y37 amylin containing N-terminal SUMO-tag [130].

Conformations described for human amylin with SUMO-tag [130] or without it [128,129] also explain the influence of amino acid substitutions found in rodent relative to human amylin to its aggregation capacity. Five substitutions in the region of 20–29 a.a. F23L, A25P, I26V, S28P, and S29P led to the destruction of two middle β-strands and protofibril interaction interface, thus preventing amyloidization of rodent amylin.

The secondary structure of in vitro formed wild-type amylin amyloid fibrils were previously determined by electron paramagnetic resonance (EPR) [131], solid-state NMR [132,133] hydrogen/deuterium exchange NMR [134] and X-ray crystallography [135] and obtained structures differed significantly from each other as well from the structures determined by cryo-EM [128,129]. Previously, the experiments were frequently conducted with non-amidated at Y37 amylin that affects the conformation of aggregated protein. Another reason for this diversity may be mentioned above structural polymorphism of in vitro aggregated amylin. The analysis of structurally distinct polymorph or the mixture of polymorphs will definitely lead to a different result. In the latter case, cryo-EM opens an opportunity to conduct separate investigation of each polymorph.

Recently, Cao and coworkers [136] attempted to elucidate T2DM-specific conformation of amylin fibrils. They investigated the structure of amylin assemblies seeded by T2DM-patient-extracted fibrils. Eight different polymorphs (four twisted and four untwisted) were identified and the structure of twisted polymorphs (TW1-TW4) was further characterized by cryo-EM (Figure 13). Among these four polymorphs, TW3 polymorph matched the structure of previously described unseeded assemblies of wild-type amylin and probably was a product of spontaneous amylin aggregation. TW1 and TW4 possessed one of two protofibril (chain b) that showed folding patterns previously described for unseeded assemblies of S20G amylin, but not usual for wild-type protein [128]. The authors considered this as a structural element derived from the pathogenic seeds [136]. Finally, TW2 represented a unique conformation that did not match neither those of TW1, TW3, TW4, nor those described for in vitro formed amylin fibrils and, thus, it is most likely represent the conformation of patient-derived amylin seeds [136].

To sum up, recent advances in structure elucidation of unseeded and seeded amylin assembles revealed the wide structural diversity of amylin fibrils. This diversity can be explained by the propensity of amylin to adopt different core folds as well as the propensity of different conformations to form various interaction interfaces [136]. Further investigations should be focused on clarifying of T2DM-specific conformation as in the only research covering this issue [136], the material from single patient has been used and pathological aggregates were amplified in seeding reaction.

## 3. Co-Pathologies: The Influence of Protein Interaction and Disease-Specific Strains of Pathological Proteins

It is known that pure amyloidoses caused by the aggregation of a particular protein are quite rare and the disease is usually characterized by the presence of co-pathologies—deposits of other protein aggregates. In this case, symptomatic disease manifests, whereas co-pathologies may be either purely coincidental or have the influence on the disease development and clinical presentation [53]. For example, tau pathology may be present early in age without any clinical symptoms but be a prerequisite for the development of PART or AD. If another specific pathogenic event occurs such as Aβ aggregation, it may remain as co-pathology, or trigger to AD-related tau-pathology when Aβ aggregation exaggerates [53].

Several hypotheses exist to account for neurogenerative disease co-pathologies including ageing [137], cross-seeding of additional proteins by misfolded amyloidogenic protein [138,139] as well as genetic risk factors such as *ApoE ε4* that increase the probability of AD manifestation [43] and probably can accelerate AD co-pathologies across other diseases. These hypotheses were tested by Robinson and co-workers [140], who investigated 766 autopsied samples representing a broad spectrum of neurodegenerative diseases, related to the aggregation of tau, Aβ, α-Syn, and TDP-43 as well as a group with minimal pathology, where no neuropathological disease was clinically diagnosed. Previous observation that tau and Aβ pathology occur with age independently of neurodegenerative diseases [36] was confirmed by the observation that both co-pathologies were common across tauopathies, synucleinopathies, TDP-43 proteinopathies as well as in minimal pathology group [140]. Groups with the highest age at death also had the highest frequencies of co-pathologies, thus confirming the hypothesis that proteinopathies are the part of normal ageing process. *ApoE ε4* has been found to be a risk factor for co-pathologies independent on neurodegenerative disease [140].

Two sets of evidences support the proteopathic seeding hypothesis. First, in AD and LBD groups, but not in the case of TDP-43 proteinopathies, more severe primary pathology exhibited higher prevalence of co-pathologies. For example, the presence of multiply co-pathologies increased from 9% in intermediate AD to 25% in high AD and from 0% for brainstem LBD to 21% for neocortical LBD [140]. Second, different tauopathies and synucleinopathies displayed distinct pattern of co-pathologies and burdens. For example, TDP-43 co-pathology was not found in PiD, whereas it was found in 24% of CBD cases. Aβ and α-Syn co-pathologies were more frequent in PSP that in PiD and CBD [140]. Thereby, these results support the hypothesis that misfolded amyloidogenic protein can cross-seed the aggregation of additional proteins. Moreover, this suggests that cross-seeding ability may be determined by the distinct conformation (strain) of pathological protein [140].

The importance of protein interaction in co-pathologies appearance may be illustrated by the relationship between AD and T2DM—two the most prevalent protein misfolding disorders. The exact mechanism of risk association between T2DM and AD is not clear but there is a set of evidence in support of direct interaction between amylin and Aβ in vivo that will be discussed in details below. From this point of view, the investigation of the interaction of pathological proteins is of particular importance in terms of disease diagnostics, disease prognosis, and the development of therapeutic approaches for disease treatment. In the next paragraphs, we review what is known to date about interaction between amyloidogenic proteins and peptides reviewed here.

## 4. Pathological Protein Interaction

During last two decades, a number of studies have been conducted to investigate pathological proteins interaction in vitro, in vivo, and in silico, and studies related to Aβ occupy the central position. For example, in a recent review, the interaction of Aβ with 16 proteins that are prone to aggregation and not related to AD was described [141]. Increased interest to Aβ is determined first of all by its involvement into AD pathological process, and also by growing interest in the relationship between T2DM and AD, as this relationship may involve the direct interaction of Aβ and amylin. Therefore, in this section, we will begin from the discussion on the interaction of Aβ with other proteins.

### 4.1. Amyloid-β and Amylin Interaction

The interaction between amylin and Aβ peptide is probably the most widely investigated example of amyloidogenic protein interaction. The reason is that apart from T2DM and AD being the two most prevalent protein misfolding disorders, one of the diseases significantly increases the risk of the other [15,16,17,18]. The exact mechanism of risk association between T2DM and AD is not clear and several hypotheses were proposed such as altered insulin signaling, impaired glycose and lipid metabolism (metabolic syndrome), and reduced Aβ clearance capacity [142]. On the other hand, there is a set of evidence in support of direct interaction between amylin and Aβ. Amylin has been found to be co-deposited together with Aβ in senile plaques of AD patients [143] and aggregated Aβ and phosphorylated tau co-localized with amylin deposits in pancreas of T2DM patients [144].

The direct interaction of Aβ and amylin has been investigated by in vitro experiments [145,146,147,148], in vivo by the analysis in yeast [149] and *Drosophila* models [150], and in transgenic mice [139], as well as in silico, by mathematical modelling using amyloid conformations described for both peptides [129,130]. A sequence alignment of amylin and Aβ40 reveals 25% of sequence identity and 50% of sequence similarity (Figure 14A). Sequence identity and similarity become even higher when amylin region 20–29 a.a. residues that are critical for aggregation (corresponds to Aβ region 25–34 a.a. residues) are taken into account (rise to 44% and 66% identity and similarity, correspondingly). The implementation of S20G mutation into amylin sequence leads to further increase in identity and similarity of amylin and Aβ to 55% and 77%, correspondingly. Taking into account the aggregation capacity of both peptides and high sequence similarity and identity, the question rises if aggregation of one of these proteins can be seeded by the fibrils of another protein. O’Nullain and co-workers [145] showed that amylin monomers can be effectively seeded by both amylin and Aβ40 fibrils. On the other hand, Aβ40 monomers were poorly seeded by amylin aggregates exhibiting 50 times lower activity compared to Aβ40 fibrils [145]. The ability of amylin aggregates to seed Aβ aggregation had been further confirmed by other researchers [139,147] and resulting fibrils were shown to be composed of both peptides [139]. There are five hot regions for Aβ40-amylin interaction: three regions (19–22 a.a., 27–32 a.a. and 35–40 a.a.) in Aβ40 and two regions (8–18 a.a. and 22–28 a.a.) in amylin (see Figure 14A and [147]).

Interestingly, amylin with N-methylated amide bonds at G24 and I26, i.e., inside the region critical for amylin aggregation and interaction with Aβ, can completely block aggregation of both Aβ40 and amylin in vitro and diminish the toxic effect of both peptide’s assemblies in cell culture [146].

Krotee and co-workers [148] investigated the structure of undecapeptides Aβ(24–34) and amylin(19–29) S20G that were derived from wild-type Aβ and S20G-amylin (Figure 14A) and found a high degree of structural similarity between backbone atoms of these fragments. Structural variants for assemblies composed of two peptides were calculated. Fibrils of both Aβ(24–34) and amylin(19–29) S20G induced amyloid formation through self- and cross-seeding and peptide inhibitors designed for one segment showed cross-efficacy for full-length Aβ and amylin and reduced cytotoxicity of both proteins in the cell culture [148].

Transgenic mice expressing both human amylin and APP exhibited exacerbated AD-like pathology compared with mice expressing human APP only or expressing human APP with streptozotocin-induced type 1 diabetes mellitus (T1DM) background [139]. Amylin colocalized with Aβ plaques in brain parenchyma deposits. The intracerebral injection of amylin aggregates led to more severe AD pathology and memory impairment in AD transgenic mice. Interestingly, amylin deposits in pancreas were substantially higher in animals overexpressing both human amylin and APP, relative to animals expressing human amylin only, suggesting that Aβ pathology may also contribute to pancreatic amylin aggregation [139]. Thereby, these data assume direct interaction of these peptides resulted in aggravation of the disease–AD in case of T2DM and, probably, vice versa. The interaction between human amylin and Aβ have been also investigated in a yeast model system [149]. Using FRET, the authors have shown that Aβ42 and amylin co-localize and interact physically in yeast.

Thus, an extensive set of evidence from in vitro and in vivo experiments supports the hypothesis of direct interaction of amylin and Aβ. Accumulated data on the peptides conformations in amyloid aggregates obtained by NMR, X-ray crystallography, EPR and especially recent advances in cryo-EM, where precise conformation of single assembly polymorph can be established, open wide perspectives in modelling of amyloid proteins interaction. Based on recently obtained cryo-EM structures aggregated wild-type- and S20G-amylin and cryo-EM structures of Aβ several models of possible amylin–Aβ interaction were developed (Figure 14B–D). The superimposition of amylin structures described by Cao and co-workers [130,136] and Röder and co-workers [129] with previously reported Aβ fibril structures revealed six different pairs of structurally similar regions in both peptides. Consensus epitopes are 14–33 a.a. residues for amylin and 16–31 a.a. residues for Aβ, which covers partially hot regions of Aβ40–amylin interaction described by Andreetto and co-workers [147] and include complete sequences of undecapeptides Aβ(24–34) and amylin(19–29) investigated by Krotee and co-workers [148].

The model proposed by Röder and co-workers [129] where S-fold of amylin is compared with LS-fold of Aβ (amylin 14–22 vs. Aβ 16–24) needs special attention. Ser20 of amylin and Glu22 of Aβ are located there in structurally similar positions. S20G substitution in amylin leads to dramatic changes in subunit structure of amylin amyloid protofibril [128]. It is suggested that the implementation of Arctic mutation (E22G) in Aβ might have analogous conformational consequences for subunit structure in Aβ protofibril [129]. The conformation of peptide in E22G-Aβ assembles has not been elucidated yet and, thus, the experimental testing of this hypothesis has to be carried out in the future.

To sum up, Aβ–amylin interaction has been extensively studied during the last two decades using various in vitro, in vivo, and in silico approaches. The capacity of amylin fibrils to seed Aβ aggregation as well as cross-seeding of amylin by Aβ assemblies have been approved experimentally. Significant progress has been reached in elucidation of near-atomic-resolution structures of amyloid fibrils of both peptides, mapping of interaction interface and analysis of the influence of this interaction on the development of T2DM and AD pathology. These results may be used for the development of new therapeutic agents for treatment of both AD and T2DM and the experimental approaches can be further applied for the analysis of interaction of other amyloidogenic proteins.

### 4.2. Amyloid-β and Tau Interaction

The interaction of Aβ and tau leads to the development of AD–unique dual proteinopathy that characterized by deposition of both proteins. Tau pathology can be met without co-existence of any Aβ pathology, for example, in PART [152]. Aβ plaques and NFTs begin in different brain regions and show different progression pattern [153], but in AD, they are converged, forming single pathologic lesion–neuritic plaque–intracellular tau inclusion together with extracellular aggregated Aβ, a distinct hallmark of AD [53]. Neuropathological data also indicate that Aβ pathology is necessary to spread tau pathology in AD out of the medial temporal lobe [153]. Tau pathology in the absence of Aβ pathology rarely exceeds NTF-Braak stage IV, while stages V/VI are present when Aβ pathology is also developed [36,152].

Pathogenic mutations in *APP*, *PSEN1,* and *PSEN2* genes lead to an increased production of Aβ and to the development of familiar AD with accumulation of both aggregated Aβ and tau. On the other hand, mutations in *MAPT* gene lead to tauopathies, but not to AD [154]. Crossing transgenic mice which develop Aβ pathology with transgenic mice demonstrating tauopathy resulted in exacerbated tau pathology in progeny [155]. In mouse primary cell culture, the treatment by exogenous Aβ oligomers induced phosphorylation of tau in AD-specific manner and its mis-sorting into dendrites [156]. Thus, the development of Aβ pathology is a prerequisite for the development or aggravation of tau pathology and further development of AD.

The regions of interaction between tau and Aβ were determined [157]. The strongest binding of tau has been shown for three regions of Aβ42: 11–16, 27–32, and 37–42 a.a. residues, and interaction regions of tau located at multiply sites, especially in exons 7 and 9. The binding enhances the phosphorylation of tau by GSK-3β and promotes Aβ nucleation into aggregates. Phosphorylated tau loses the affinity to Aβ, which results in dissociation of the complex. Phosphorylation of T212, S214, S356, and S396 in tau completely blocked Aβ42 binding, phosphorylation of T217 and T231 decreased the binding significantly, whereas phosphorylation of T181, S202, and T205 had no effect [157]. Complexes of tau and Aβ were detected in soluble extracts from AD and control brain tissues and double immunostaining of AD brain tissue recognized phosphorylated tau and Aβ to form separate insoluble complexes within the same neurons and their processes [157]. The authors hypothesized that in AD, an initial step in the pathogenesis may be the intracellular binding of soluble Aβ to soluble non-phosphorylated tau, thus promoting tau phosphorylation and Aβ nucleation [157]. This hypothesis explains the convergence in the development of tau and Aβ pathology as well as Aβ-stimulated aggravation of tau pathology in AD. It is also in accordance with the recent data on disease-specific strains of pathological tau. The interaction of Aβ and tau leads to the implementation of specific PTMs into tau, for instance, by the phosphorylation of Ser256 with GSK-3β in Aβ-tau complex. These specific PTMs may divert phosphorylated tau toward the aggregation in AD-specific amyloid conformation.

### 4.3. Amyloid-β and TDP-43 Interaction

TDP-43 pathology is detected in AD brains in 20–40% of AD cases [140,158] with higher prevalence of TPD-43 pathology in severe AD [140]. TDP-43 pathology has been shown to be associated with more extended brain atrophy and greater memory loss as well as with the severity of AD pathology, namely higher Braak NFT stages and Thal amyloid phases [159]. On the other hand, in case of ALS and FTLD-TDP, Aβ pathology was discovered in 36% and 42%, correspondingly, and severity of TDP-43-related disease did not correlate with the presence Aβ pathology [140].

A triple increase in full-length TDP-43 level has been found in brain cortex of late-stage AD patients relative to non-AD control [160]. Similar to the pathological changes seen in human AD brain, the lentiviral expression of Aβ42 in the rat motor cortex led to the increase in TDP-43 level, its phosphorylation and accumulation in cytosol. The injection of lentiviral parkin that stimulates intracellular Aβ42 clearance prevented TDP-43 pathology [160]. Lentiviral expression of α-Syn did not affect TDP-43 and parkin co-expression with Aβ42 did not affect the development of tau pathology. These results suggest that TDP-43 pathology is triggered specifically by elevated level of Aβ42 and by the mechanism independent on tau [160].

The overproduction of TDP-43 in cortex and hippocampus of APP/PS1 mouse that possesses mutant *APP* and *PSEN1* genes increased abnormal tau aggregation and decreased Aβ plaque deposition with a simultaneous increase in endosomal/lysosomal APP localization [161]. Additionally, the elevation of TDP-43 level decreased the activity of calcineurin, a phosphatase for TDP-43, thus promoting the shift toward to TDP-43 phosphorylation [161]. Interestingly, knockout of TDP-43 in forebrain of APP/PS1 mouse also decreased Aβ plaque deposition, but led to neurodegeneration and accumulation of prefibrillar oligomeric Aβ assemblies [162]. Specific knockout of TDP-43 in microglia led to acceleration of its phagocytic activity that may explain decreased Aβ plaque formation due to enhanced Aβ clearance [163]. At the same time, microglial knockout of TDP-43 led to synaptic loss [163]. Thus, both elevation and decrease in TDP-43 level may contribute to AD pathological process by the influence to tau phosphorylation, Aβ aggregation state, Aβ clearance, and neurodegeneration.

Recombinant full-length TDP-43 inhibited in vitro Aβ fibrillization at initial and oligomeric stage. It was shown that TDP-43 binds Aβ through RBD domain(s), N-terminus of TDP-43 is critical for inhibition of Aβ fibrillization, while C-terminus is not necessary for Aβ-TDP-43 interaction [164]. The presence of TDP-43 fragments containing both N-terminus and RBD in the reaction mixture triggered Aβ aggregation from fibrillization to predominant formation of oligomers [164]. The intrahippocampal injection of TDP-43 led to increased level of Aβ oligomers, inflammation, and exacerbation of AD-like pathology [164]. At the same time, TDP-43 can form in vitro structurally stable spherical oligomers that can be recognized with an anti-amyloid oligomer-specific antibody [84]. These oligomers were capable of cross-seeding Aβ to form amyloid oligomers [84]. Such oligomers were extracted from the brain of transgenic TDP-43 mice and from the brain samples of AD-, ALS-, and FTLD-TDP patients [84,87]. In the brain of AD patients and of APP/PS1 mouse, TDP-43 oligomers mainly colocalized with intraneuronal Aβ and partly with amyloid plaque [164]. On the other hand, Aβ42 oligomers can seed recombinant TDP-43 protein and accelerate the formation of TDP-43 oligomers [165]. Importantly, Aβ42 fibril seeds were able to accelerate formation of Aβ42 fibrils, but not TDP-43 fibrils. Additionally, oligomeric structures composed of TDP-43 and Aβ or αSyn were shown to be present in AD-brain parenchyma and vasculature [165]. Similarly with tau-associated pathologies, when the pattern of tau pathology distribution is different between AD and other tauopathies, spreading of TDP-43 pathology in AD differs from that in FTLD and ALS [166,167] and detailed staging scheme for classifying TDP-43 pathology in AD has been recently developed [167].

To sum up, TDP-43 and Aβ can interact and this interaction has been confirmed in the experiments in vitro and in vivo. TDP-43 pathology may accompany AD and aggravate disease progression by the mechanism that may involve enhanced tau phosphorylation and Aβ oligomer production. On the other hand, Aβ and tau oligomers may affect TDP-43 subcellular distribution and aggregation state, thus governing the spreading of TDP-43 pathology in AD-specific manner.

### 4.4. Amyloid-β and α-Syn Interaction

Up to 60% of AD cases exhibit significant LB pathology in addition to Aβ plaques and NFT [140,168], whereas in a group with minimal pathology, it was recognized as being 10 times less frequent [140]. Across LBD patients, Aβ pathology prevalence was around 50% that was similar with aged control group with minimal pathology, except severe LBD cases with LB presented in neocortical region, where Aβ deposits were recognized in 80% of cases [140]. As levels of AD neuropathology increased, cerebral α-Syn scores were also higher, and the interval between the onset of motor and dementia symptoms as well as disease duration were shorter [169]. The most severe cases can be characterized as mixed AD/LB dementia since the severity of either AD or LB pathology alone would be sufficient for clinical dementia. In these cases, one pathology may be primary and drive neurodegeneration and the development of the other pathology as concomitant [169]. Tight overlapping of AD and LB pathologies suggests the possible interaction of Aβ and α-Syn.

Among transgenic mice with neuronal expression of Aβ, α-Syn, or both, doubly transgenic mice resembled the phenotype of LB variant of AD [170]. These mice were characterized by more extensive α-Syn neuronal pathology and developed severe learning, memory, and motor deficit earlier than α-Syn singly transgenic mice [170]. In vitro experiments showed that α-Syn can form oligomers and higher molecular weight aggregates in the presence of equimolar amount of Aβ and, thus, enhanced α-Syn accumulation in the brain and neuronal deficit in doubly transgenic mice was explained by the interaction of Aβ and α-Syn that promotes the aggregation of the later protein [170].

The capacity of these two proteins to interact has been studied in vitro using multidimensional NMR spectroscopy [171]. Aβ and α-Syn interacted in aqueous solution with SDS that mimic membrane environment, as was evident by the alteration of chemical shift of amide protons corresponding to a.a. residues 3–37 of Aβ40 and 10–140 of α-Syn [171]. In case of Aβ42, the interaction was non-specific and covered all the residues of the peptide [171]. It has been proposed that Aβ–α-Syn interaction take place at the surface of synaptic membrane, and it results in conformational changes of α-Syn that initiate its aggregation [171]. On the other hand, the modelling of the interaction between Aβ oligomers and oligomers formed by NAC region of α-Syn demonstrated that NAC oligomers induced formation of new β-strands in Aβ42, thus leading to new, more stable and compact cross-β structure of Aβ42 oligomers in heterologous assemblies [172].

It has been shown that Aβ and α-Syn are co-immunoprecipitated from the brain samples of AD/PD patients and transgenic mice [173]. Additionally, these proteins were able to form hybrid, ring-shaped pore-like oligomers [173]. HEK293 cell culture overproducing α-Syn displayed calcium influx consistent with the formation of ion channels after the exposure by exogenous Aβ [173], thus demonstrating a potential detrimental effect of Aβ–α-Syn interaction on cell membrane integrity. Ono and co-workers [174] showed that both fibrillar and oligomer seeds of Aβ can induce aggregation of αSyn and vice versa. α-Syn seeds were more potent than those of Aβ in seeding of both α-Syn and Aβ, and fibrillar assemblies of both proteins seeded the aggregation more effectively than oligomers [174]. The propensity of Aβ42 oligomers to seed α-Syn oligomerization in vitro has been confirmed later and α-Syn oligomers were histochemically detected in human AD brain samples [165]. Importantly, in that study, Aβ42 fibrils could not seed α-Syn aggregation in vitro and it was proposed that oligomer rather than fibrillar Aβ seeds act as a template for α-Syn and other proteins, namely PrP and TDP-43 [165].

To sum up, Aβ and α-Syn can interact and synergistically aggravate each other’s pathology in AD and LBD. For this reason, the interaction of Aβ and α-Syn is an important target for the development of novel therapeutic approaches for treatment of both AD and LBD.

### 4.5. Tau and α-Syn Interaction

The interaction of α-Syn and tau has been firstly described by Jensen and co-workers [175], who showed that α-Syn binds tau and stimulates tau phosphorylation by protein kinase A. It was proposed that α-Syn modulates the phosphorylation of soluble tau and thereby indirectly affects the stability of axonal microtubules [175]. At the same time, tau and α-Syn aggregated together in the same deposits [176,177,178,179]. Moreover, some inclusions were comprised of both proteins with almost complete spatial disparity [177].

Certain mutations of *SNCA* or *MAPT* gene were found to result in elevation of tau- or α-Syn pathology, correspondingly, giving additional evidence of possible α-Syn–tau interaction. For example, α-Syn mutation A53T that is known to increase aggregation propensity of α-Syn [180,181] results in widespread of both α-Syn and tau inclusions in the brain [177]. *MAPT H1* haplotype was identified as genetic risk factor of PD [182] and DLB [183]. Additionally, a significant association with DLB and LBD was shown for point mutation A152T in tau [184].

Experiments in vitro displayed that α-Syn induced fibrillization of tau and that coincubation of α-Syn and tau synergistically promotes fibrillization of both proteins [185]. All six major tau isoforms were prone to aggregation in presence of α-Syn. Both proteins formed fibrils and most of them were homopolymers with only rare α-Syn-formed fibrils also displayed sporadic tau-immunoreactivity. Moreover, some fibrils were found to possess spatially separate domains with labeling by either αSyn, or tau antibodies, suggesting that these fibrils resulted from the end-to-end annealing of filaments formed entirely by either tau, or α-Syn [185]. The co-occurrence of α-Syn and tau inclusions were shown in single transgenic mice expressing human A53T α-Syn and in double transgenic mice expressing wild type α-Syn and P301L tau in oligodendrocytes. In both cases, apart from inclusions composed of tau or α-Syn alone, there were those formed by both proteins [185].

Despite cross-seeding of tau and α-Syn being clearly described, it was not uniformly reproduced in further studies [186]. This phenomenon was explained when two polymorphs (strains) of aggregated α-Syn with distinct propensity to seed aggregation of tau were isolated [187]. The assemblies of two polymorphs did not differ in morphology and in physio-chemical properties and represented almost the same pattern of proteolytic fragments after digestion by proteinase K [187]. Nevertheless, experiments with neuronal cell culture and transgenic mice displayed significant difference in biological activity between two polymorphs. Fibrils of the first polymorph were toxic for cell culture and induced extensive α-Syn pathology in the absence of any tau aggregation. Oppositely, fibrils of second polymorph had no negative impact on cell survival and induced aggregation of both α-Syn and tau with high proportion of co-localization. Similar results were obtained with transgenic mice [187]. Importantly, α-Syn assemblies prepared from N-terminal-truncated forms were incapable to seed tau; thus, N-terminal sequence (1–58 a.a.) appears to be necessary to form cross-seeding complement for α-Syn–tau interaction [187]. On the other hand, when α-Syn promotes tau phosphorylation by protein kinase A, C-terminus of α-Syn is involved into interaction [175]. Additionally, Guo and co-workers [187] showed, that pathological α-Syn derived from PD dementia and PD/AD dementia brain may be conformationally different. In the previous section, we described recent advances in elucidation of disease-specific conformation of pathological α-Syn but in correspondence to different diseases. This result suggests multiple amyloid conformations of the protein within one disease.

It has been also shown that tau can form SDS-resistant oligomers in vitro and oligomerization is synergistically enhanced by physiologically relevant concentrations of Al^3+^ and tau phosphorylation by GSK-3β [188]. Moreover, trivalent ions such as Al^3+^ and Fe^3+^, organic solvents and phosphorylation of tau facilitated co-oligomerization of tau with α-Syn as well as recruitment of α-Syn in pre-formed tau oligomers [188]. Co-oligomerization of amyloidogenic proteins that was also described for α-Syn–TDP-43, Aβ–α-Syn and Aβ–TDP-43 interaction [165,173] may reflect common pathological mechanism explaining cross-seeding phenomenon and coincidence of different pathologies in one disease.

To sum up, α-Syn and tau can interact in multiple ways. Soluble proteins can synergistically promote fibrillization of both proteins. α-Syn assemblies can seed aggregation of tau and seeding activity of α-Syn aggregates is determined by the conformation of α-Syn. Additionally, α-Syn and tau can form oligomers composed of both proteins, thus representing the features shown for interaction of other proteins.

### 4.6. Tau and TDP-43 Interaction

Among various tauopathies, TDP-43 inclusions were found in AD, CBD, and PSP, but not in PiD [79,140,189,190,191]. In AD, despite a subset of TDP-43-positive inclusions co-existing with NFTs in the same neurons, a double-immunofluorescent analysis assay did not show any co-localization of these structures [79,189]. In CBD, TDP-43-positive inclusions showed partial but not complete overlap with tau immunoreactivity [189]. In PSP, the co-localization existed, but it was region-specific and presented in amygdala, but not in hippocampal gyrus [190]. Also, immunoblotting demonstrated the appearance of smear corresponded to high molecular weight TDP-43 aggregates in the TDP-43-positive AD, CBD, and PSP cases [189,190].

It has been shown that tau can modulate cellular localization and oligomerization of TDP-43 [87]. Exogenous treatment of cell culture with soluble human recombinant tau oligomers increased the level of both phosphorylated and non-phosphorylated TDP-43 monomers in nucleus, as well as the amount of TDP-43 oligomers in cytoplasm. At the same time, tau oligomers induced redistribution of TDP-43 oligomers from low-molecular-weight (150 kDa) to high-molecular-weight fraction (>250 kDa). The overproduction of tau in cell culture did not affect *TARDBP* gene expression, but decreased of TDP-43 oligomers in the nucleus with a simultaneous increase in TDP-43 oligomers in the cytoplasm [87]. TDP-43 oligomers of various pathologic origin (AD, ALS and FTLD-TDP) extracted from the brain samples were able to seed aggregation of recombinant tau in vitro. Nevertheless, the treatment of aggregated tau with proteinase K and subsequent PAGE did not show any disease-specific profile of proteolytic fragments [87]. Despite the ability of TDP-43 oligomers to seed oligomerization of tau, immunofluorescent analysis revealed poor co-localization of TPD-43 oligomers and tau oligomers in brain tissues from AD patients [87].

To sum up, tau can modulate oligomerization state and cellular localization of TDP-43, whereas amyloid oligomers of TDP-43 can seed soluble tau in vitro. However, despite it being supposed that TDP-43 and tau may influence each other’s pathological progression [77], this preposition has to be more extensively checked. Additional studies are required to elucidate the relationship between tau and TDP-43.

### 4.7. Tau and Amylin Interaction

Tight interdependence between AD and T2DM discussed above in terms of Aβ and amylin interaction raises an additional question if diabetes may also affect the development or aggravation of tau pathology in AD and other tauopathies, and if tau and amylin may directly interact or not. Several studies displayed enhanced tau phosphorylation in the brain under T1DM [192,193,194,195,196,197] as well as under T2DM [192,196] condition. In the T1DM model, wild-type tau was hyperphosphorylated, but not aggregated, whereas P301L mutant tau formed massive NFTs [195]. AD-specific pathology developed in T2DM, but not in the T1DM model [193,194].

The mechanism of tau phosphorylation in diabetes involves altered insulin signaling as a result of insulin tolerance. Binding of insulin to insulin receptor activates Akt pathway that results in phosphorylation of GSK-3β kinase [198]. This kinase is able to phosphorylate tau and it is maintained inactive being phosphorylated. Insulin resistance leads to disruption of Akt pathway, dephosphorylation of GSK-3β, and hence, to its activation with subsequent hyperphosphorylation of tau [198]. Additionally, diabetes results in inhibition of brain protein phosphatase 2A that is capable of dephosphorylating tau at multiply residues [192].

El Khoury and co-workers [199] have analyzed more than 30 studies reporting brain tau phosphorylation in a mouse or rat model of T1DM and T2DM and found that insulin disfunction can promote tau hyperphosphorylation and pathology, both directly and indirectly, through hypothermia. Nevertheless, Zhang and co-workers [200] have recently reported accelerated aggregation of recombinant K18 tau fragment in vitro in the presence of soluble or fibrillar amylin. Compared with heparin-induced K18-tau fibrils, heterological fibrils differed in morphology, displayed altered profile of proteolytic fragments after PK digestion, and demonstrated enhanced seeding activity and cytotoxicity in vitro [200]. The intracerebral injection of synthetic amylin fibrils initiated tauopathy in the brain of tau-P301S transgenic mice [200]. Despite the shown possibility of amylin assemblies to seed tau aggregation in vitro and induce tau pathology in vivo [200], we are rather skeptical about the existence of such a mechanism in AD for two reasons. First of all, both T1DM and T2DM conditions similarly increased tau phosphorylation and this effect was mediated by kinase activity through impaired insulin signaling and hypothermia. Secondly, we did not find any published evidence on the association of T2DM with any of the tauopathies other than AD. Thus, we suppose that even if there is a direct interaction between tau and amylin, there is currently no information that such an interaction can influence the onset and development of tau-related pathologies.

### 4.8. α-Syn and TDP-43 Interaction

The evidence on the interaction between α-Syn and TDP-43 is restricted. The TDP-43 proteinopathies, both ALS and FLTD-TDP, demonstrated the elevation of α-Syn co-pathology with the severity of the main disease burden [140]. On the other hand, different synucleinopathies demonstrated differing prevalence of TDP-43 co-pathologies. In LBD, the prevalence of TDP-43 co-pathology increased with the severity of disease being close to zero among the patients with brainstem LBD and exceeding 20% among the group with neocortical LBD showing similar dynamics with the prevalence of Aβ co-pathology [140]. In an AD/LB dementia group of patients, TDP-43 co-pathology was more frequent than in pure LBD group, but less prevalent than among pure AD patients [201]. In MSA, the prevalence of TDP-43 pathology was higher than in the control group, but lower than in case of limbic or neocortical LBD [140].

α-Syn and TDP-43 can form oligomers composed of molecules of both proteins and the presence of these aggregates was shown in AD brain [165]. The presence of chimeric α-Syn–TDP-43 oligomers similar with those described for Aβ–α-Syn [173] and Aβ–TDP-43 [165] means that cross-seeding mechanism may exist for α-Syn–TDP-43 interaction. This may partially explain increased prevalence of TDP-43 co-pathology with growing severity of LBD. On the other hand, it may be also explained by simultaneous expansion of Aβ co-pathology and further Aβ-driven aggregation of TDP-43.

To sum up, the data on α-Syn–TDP-43 interaction collected to date are rather incomplete. Additional investigations have to be conducted to understand the nature of α-Syn–TDP-43 interaction and its possible influence on the development of α-Syn- or TDP-43-related proteinopathies as separate diseases or as co-pathologies.

### 4.9. α-Syn and Amylin Interaction

The data accumulated to date suggest that T2DM is associated with an increased risk of PD [19,20]. PD patients with dementia are two times more likely to have insulin resistance than patients with PD [202], suggesting diabetic symptoms may aggravate the progression of PD. Experiments in vitro displayed that whereas both amylin and α-Syn alone can form amyloids, mixing of amylin and α-Syn monomers results in coaggregation that is faster than either protein aggregates alone [203]. Human α-Syn monomers promoted human amylin, but not non-amyloidogenic rat amylin aggregation in vitro in dose-dependent manner and resulted assemblies were composed of both proteins, suggesting that under these conditions α-Syn and amylin form hybrid amyloid fibrils [204]. Interestingly, preformed seeds and fibrils of α-Syn were not able to promote amylin aggregation [204], whereas pre-formed amylin amyloid accelerated α-Syn fibrillization [203]. Importantly, in contrast to mature amylin, pro-amylin assemblies inhibited rather than promoted α-Syn aggregation [203]. The modelling of the interaction between amylin oligomers and oligomers formed by NAC fragment of α-Syn displayed that structural features of NAC core in NAC-amylin hetero-oligomers were not affected, whereas amylin oligomers acquired more stable conformation [205]. Similar stabilization was also described for Aβ involved into heterologous NAC-Aβ oligomers [170].

Cytoplasmic phosphorylated α-Syn deposits were found in pancreatic β-cells of patients with PD, DLB, and LBD [206], as well as in samples from patients with normal neuropathological examination, but suffering from T2DM [204,206]. Moreover, every sixth patients from control group (T2DM- and neuropathologically negative) were found to possess phosphorylated α-Syn inclusions in pancreas [206]. Additionally, α-Syn was found to be a component of amyloid extracted from pancreas of transgenic mice producing human amylin [204]. Tail-vein injection of α-Syn in mice expressing human amylin enhanced amylin deposition in pancreas, whereas amyloid formation in β-cells was reduced in transgenic mice expressing human amylin on *Scna*^−/−^ background [204].

To sum up, there is strong evidence from in vitro, in vivo, and in silico studies that α-Syn and amylin can directly interact. Importantly, amylin deposits in the brain were found to be associated with T2DM [103], or with AD [143]. In the latter case, amylin and α-Syn are presented in amyloid plaques together with Aβ, but their colocalization seems to be determined by separate interaction of each protein with Aβ rather than by direct interaction of α-Syn and amylin. This preposition is supported by the fact that colocalization of α-Syn and amylin in the brain of patients with synucleinopathies has not been described yet. Thus, unlike Aβ–amylin interaction in AD and T2DM, where proteins can interact either in the brain or in the pancreas and deposits composed of Aβ and amylin can be present in both organs, α-Syn–amylin interaction seems to take place predominantly in islets β-cells.

### 4.10. TDP-43 and Amylin Interaction

The interaction of TDP-43 and amylin has not been analyzed yet. Nevertheless, recent studies discovered a negative relationship between T2DM and ALS [207,208,209,210]. This association was strong for non-insulin-dependent diabetes, but not for insulin-dependent diabetes [207]. It is known that both ALS and frontotemporal dementia are the factors decreasing the risk of cardiovascular diseases [210,211]. On the other hand, ALS patients are hypermetabolic and have impaired glucose tolerance [212,213]. In case of T2DM and ALS, one of the diseases may induce metabolic changes that compensate the risk of the another resulting in negative relationship between these pathologies.

## 5. Conclusions

The interaction between Aβ, tau, α-Syn, TDP-43, and amylin reviewed above are summarized in Table 1.

Pathological protein aggregation seems to be the part of natural aging process, and the age when this process becomes recognizable depends on the protein itself, genetic background, and concomitant diseases and co-pathologies. The interaction of pathogenic proteins also plays an important role here, as they may lead to the development of a specific disease (for instance, interaction of Aβ and tau results in AD) or to the propagation of concomitant pathologies that aggravate the progression of the main disease (TDP-43 or α-Syn pathology in AD or Aβ pathology in PD), or to the appearance of another distinct disease in addition to the main one (AD at the background of T2DM and vice versa). This is why understanding of pathological proteins interaction is important for understanding of disease nature and for the prognosis of concomitant pathologies.

The interaction may lead to direct transfer of amyloid conformation from amyloid fibrils or oligomers of one protein to monomers of another protein, as it takes place in cross-seeding (interaction of Aβ and amylin, or Aβ and TDP-43). Alternatively, the interaction may promote templating-appearance of newly formed seeds (Aβ interacting with tau, α-Syn or amylin) or implementation of new PTMs such as phosphorylation that significantly increase aggregation propensity of the protein (tau interacting with Aβ or α-Syn). Moreover, both mechanisms can be potentially realized, as has been described for α-Syn–amylin, α-Syn–Tau, or α-Syn–Aβ interaction.

It is necessary to point out that pathological proteins are not necessarily interacting as was shown for tau–amylin and TDP-43–amylin. In this case, the development of concomitant pathologies may be due to physiological and biochemical disturbances appearing during main disease progression. These disturbances affect the risk of other disease progression. For example, both AD and T2DM are characterized by insulin and glucose tolerance, slowing down the metabolism, hypertension and, thus, one disease increases the risk of the another. On the other hand, ALS is characterized by the opposite physiological changes and in case of ALS and T2DM, the risks of one disease in case of the another is lower compared to the healthy control. Another example is the influence of T2DM on tau pathologies. Despite the recently described possibility of tau–amylin interaction, hypothermia that is the consequence of T2DM may make more significant contribution to tau phosphorylation and subsequent development of tau pathology.

To sum up, recently discovered disease-specific co-pathology profile supports the hypothesis about disease-specific amyloid polymorphs. This hypothesis has been strongly approved by the result of recent cryo-EM investigations where the structure of amyloid fibrils formed by various proteins, derived from in vitro reaction or from biological samples was elucidated with near-atomic resolution. Cryo-EM analysis revealed amazing diversity of amyloid conformations and fibril architectures that can be formed even by the same protein. Described structures allows clearly explain physio-chemical properties of fibrils such as affinity to amyloid-specific dyes, the influence of point mutations on exact amyloid conformation, thus explaining their pathogenic or protective role. The comparative analysis of structures formed by different proteins opens a perspective for elucidation of the regions responsible for protein interaction, as has been shown in case of the pair amylin–Aβ. Despite recent advances in elucidation of pathological protein interaction and the impact of this interaction to the progression of the main disease and development of co-pathologies, these data are rather incomplete. The data in Table 1 cover the results of in vitro, in vivo, and in silico experiments, where in vitro-formed assemblies and truncated forms of proteins were used in most cases. Taking into account observed structural differences between in vitro-formed and ex vivo-extracted protein assembles, further investigation should be focused on the elucidation of disease-specific amyloid conformation and analysis of their interactome. The establishment of clearer relationship between conformation and disease and conformation and co-pathologies should help in the development of novel conformation-specific diagnostic dyes as well as novel pharmaceutical substances that may affect the development disease and corresponding co-pathologies by modification of protein–protein interaction.

## Figures and Tables

**Figure 1 life-13-01954-f001:**
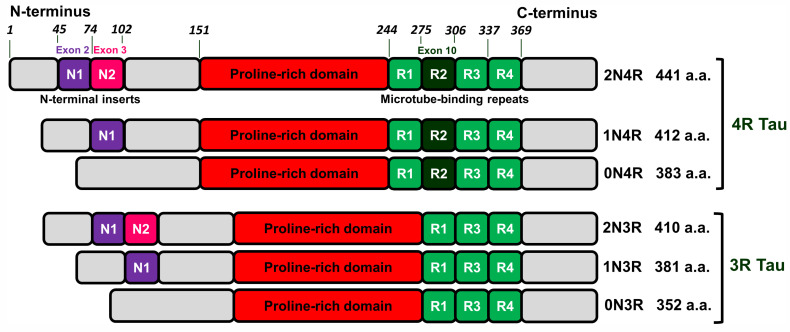
Schematic representation of human tau isoforms. Tau proteins consist of four domains: N-terminus (grey), proline-rich domain (red), microtube-binding repeats (light/dark green), and C-terminus (grey). Alternative splicing of exon 10 (dark green) gives rise tau isoforms with three or four microtube-bindings repeats named 3R and 4R isoforms. Alternative splicing of exons 2 (purple) and 3 (pink) generates isoforms with no (0N), one (1N) or two (2N) N-terminal inserts that in combination with 3R and 4R varieties gives rise to 6 isoforms 0N3R, 1N3R, 2N3R, 0N4R, 1N4R and 2N4R.

**Figure 2 life-13-01954-f002:**
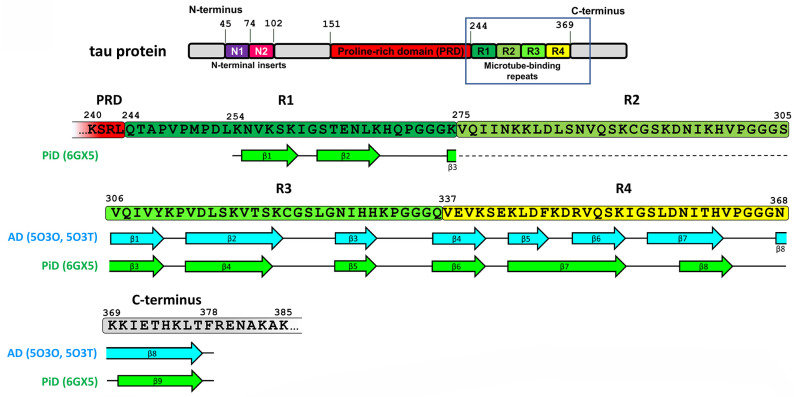
Sequence alignment of tau protein region involved in amyloid core formation with β-strand regions observed in various disease-specific tau conformation. The location of β-strands for AD fold is shown as represented in Figure 3d from [32], and for PiD fold—as in Figure 2b from [25]. The absence of R2 domain in 3R tau forming PiD-specific fold is shown by dotted line.

**Figure 3 life-13-01954-f003:**
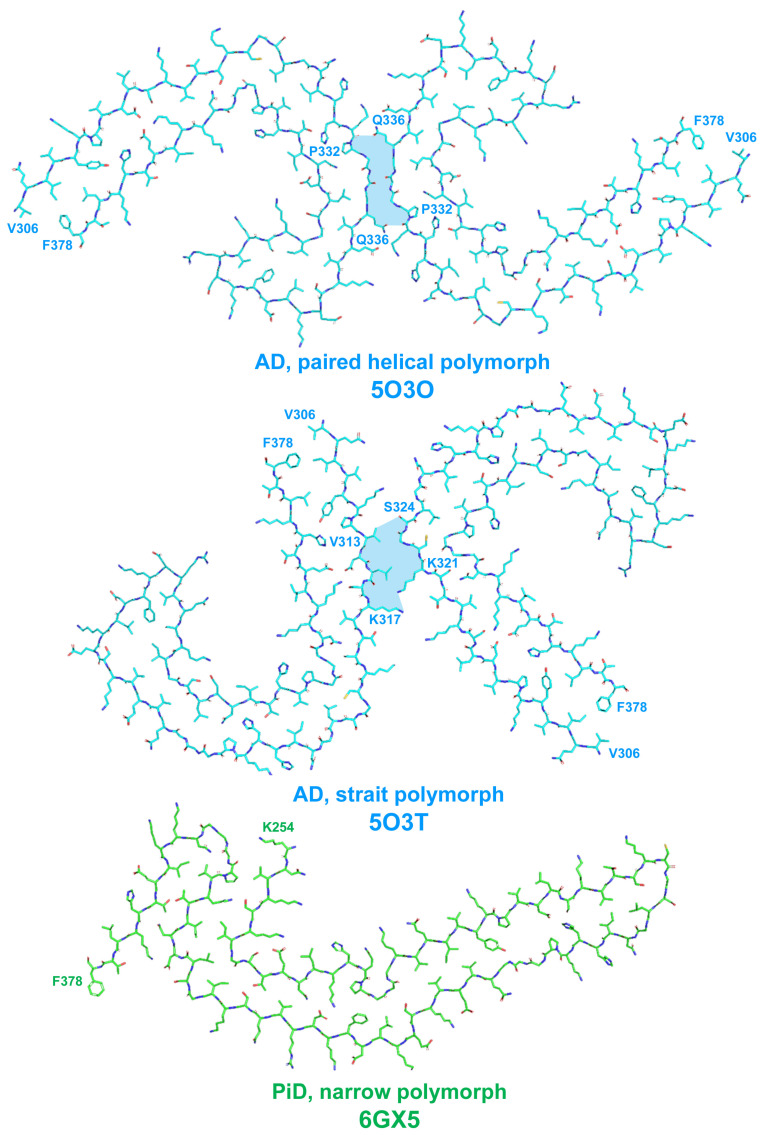
Schematic representation of disease-specific folds of tau protein. Here and on the subsequent figures, the structures are shown as represented in RCSB Protein Structure Database (https://www.rcsb.org/ accessed on 6 August 2023); RCSB PSD reference code for each structure is specified. Amino acid residues are numbered based on 2N4R human tau isoform. Inter-protofilament interface in AD-specific polymorphs is shown by light-blue filling of inter-protofilament space.

**Figure 4 life-13-01954-f004:**
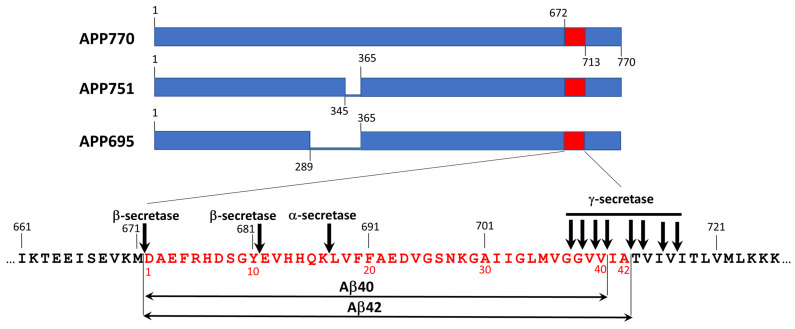
The origin of amyloid β peptides. Amyloid β peptides originate from proteolytic cleavage of amyloid precursor protein (APP) that is present in the cell in three different isoforms (APP695, APP751, and APP770) by β- and γ-secretases. APP cleavage by α-secretase prevent formation of aggregation-prone fragments and, thus, is protective. β- and γ-secretases can cleave APP at different sites producing a series of proteolytic fragments composed of 36–46 a.a. residues with Aβ40 and Aβ42 are the most abundant peptides. Expansion represents partial sequence of APP, covering Ab42 (shown in red). Cleavage site for α-, β-, and γ-secretases are indicated by arrows. All APP sequences are numbered according to the full-length APP (APP770) sequence. The sequence at the expansion is numbered according to APP770 on the top of sequence (black numbers) and according to Ab42 on the bottom of the sequence (red numbers).

**Figure 5 life-13-01954-f005:**
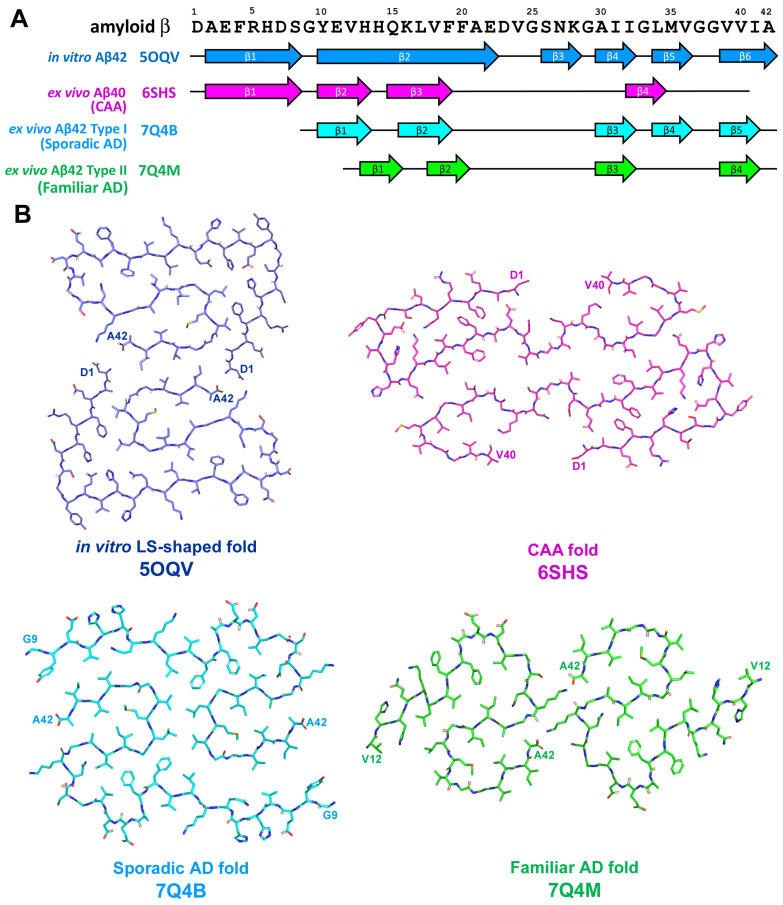
(**A**) Sequence alignment of amyloid β peptide with β-strand regions observed in various in vitro-derived and ex vivo extracted Aβ assemblies. The location of β-strands for in vitro Aβ42 fold is shown as represented on Figure 3B from [50], for ex vivo Aβ40 CAA-specific fold–as on Figure 3b from [52] and for ex vivo AD-specific Aβ42 folds type I and II–as on Figure 2D,E from [51], correspondingly. (**B**) Schematic representation of in vitro-derived and ex vivo disease-specific folds of amyloid β peptide.

**Figure 6 life-13-01954-f006:**
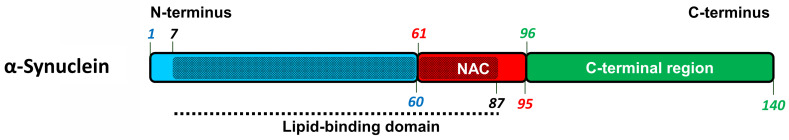
The structure of α-synuclein protein. The location of lipid-binding domain (7–87 a.a.) is mapped by shading and black dotted line. NAC (61–95 a.a.)–non-amyloid component.

**Figure 7 life-13-01954-f007:**
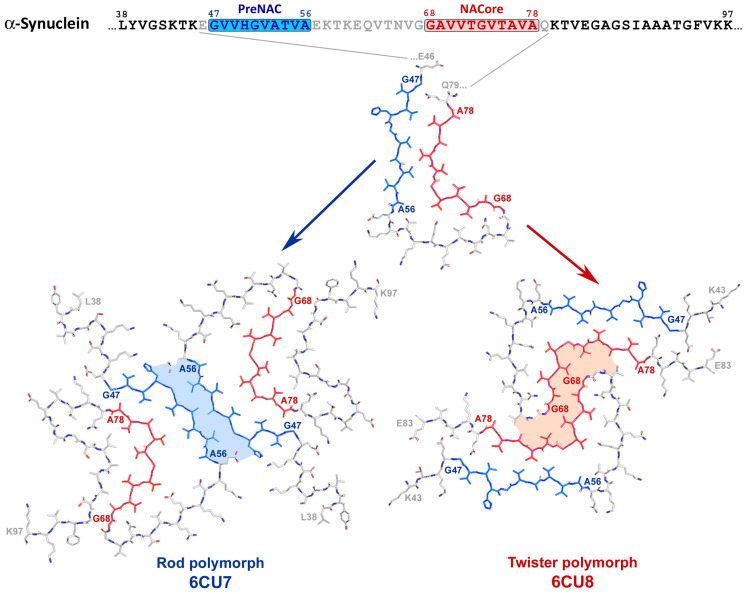
The structure of rod and twister polymorphs of in vitro assembled full-length α-synuclein fibrils. PreNAC (47–56 a.a.) and NACore (68–78 a.a.) regions critical for aggregation of α-Syn are highlighted in blue and red, correspondingly. Protofibrils of both polymorphs shared common kernel structure covering PreNAC, NACore, and the region between them. PreNAC region provides the protofibril interaction in rod polymorph, whereas the NACore site is responsible for this interaction in twister polymorph. Inter-protofilament interface in polymorphs is shown by light-blue and pink filling of inter-protofilament space in rod and twister polymorph, correspondingly. Adopted from [66].

**Figure 8 life-13-01954-f008:**
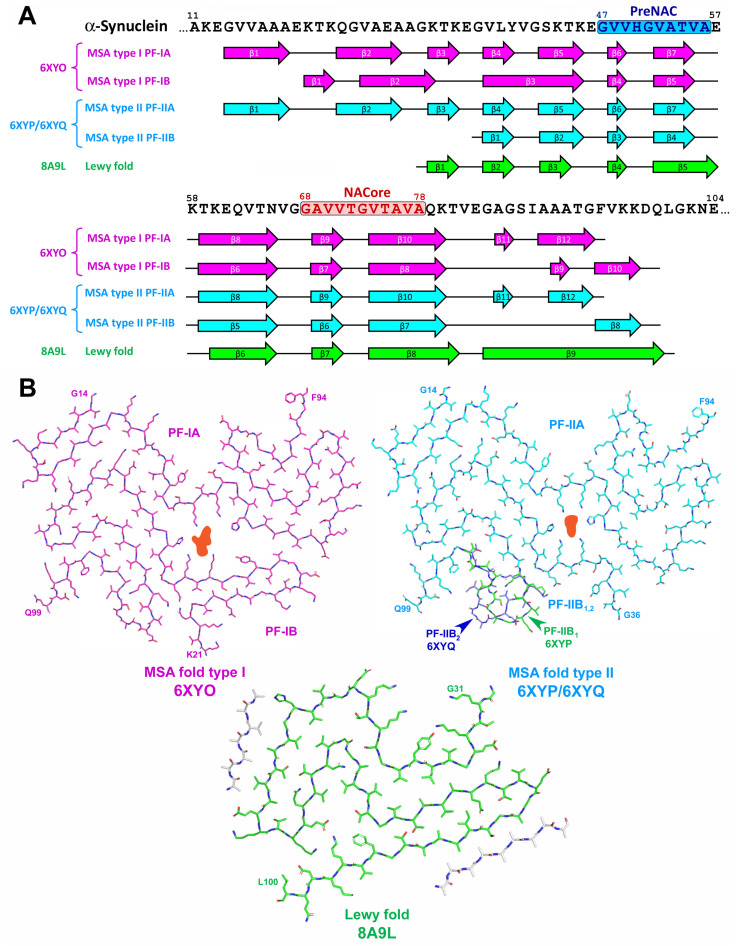
The structure of disease-specific α-synuclein assemblies. MSA-derived aggregates are represented by two types of filaments (folds type I and type II), each of them composed of two asymmetrically folded protofilaments (PF-IA and PF-IB for MSA fold type I and PF-IIA and PF-IIB for MSA fold type II). (**A**) Sequence alignment of α-Syn with β-strand regions observed in ex vivo extracted α-Syn assemblies. PreNAC and NACore regions are highlighted. The location of β-strands for MSA folds type I and type II is shown as represented on the Extended Data Figure 5 from [72], for Lewy fold–as on Figure 2 from [75]. (**B**) Schematic representation of ex vivo extracted disease-specific folds of α-Syn. Structure variation for PF-IIB is shown by green (PF-IIB_1_) and dark blue (PF-IIB_2_). The non-proteinaceous density at the protofilament interface of MSA folds is shown in light red. Unidentified protein fragments in Lewy fold are shown in grey.

**Figure 9 life-13-01954-f009:**
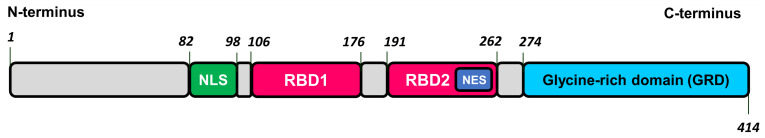
The structure of transactive response DNA binding protein of 43 kDa (TDP-43). NLS–nuclear localization sequence, RBD1 and RBD2–RNA-binding domains, NES–nuclear export sequence.

**Figure 10 life-13-01954-f010:**
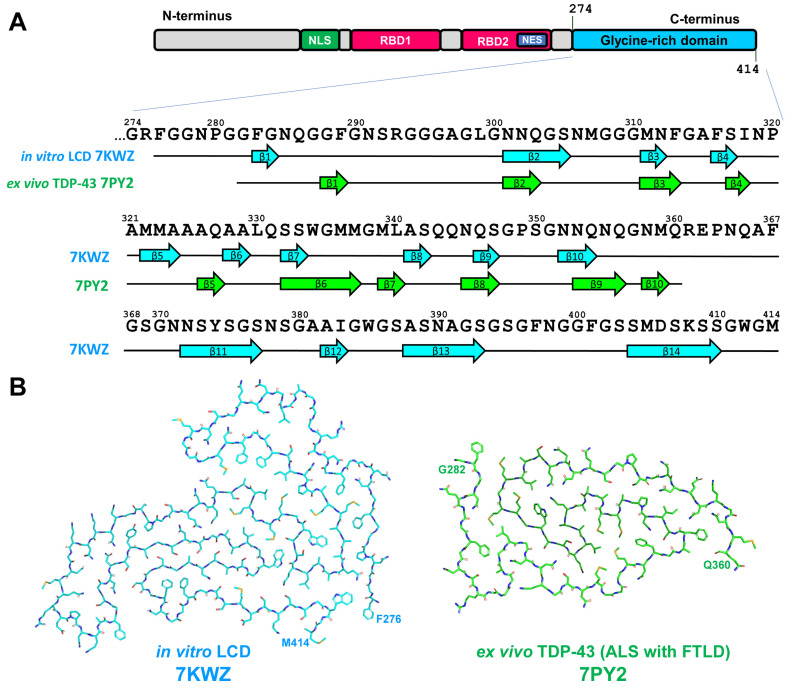
(**A**) Sequence alignment of TDP-43 glycine-rich low complexity domain (LCD) 274–414 a.a. with β-strand regions observed in in vitro-aggregated LCD and in ex vivo extracted TDP-43 assemblies. The location of β-strands for in vitro-aggregated LCD is shown as represented on Figure 1d from [89], for ex vivo extracted TDP-43 assemblies (ALS with FTLD)—as in Extended Figure 4a from [90]. (**B**) Schematic representation of in vitro-derived and ex vivo ALS-with-FTLD-specific folds of TDP-43 LCD.

**Figure 11 life-13-01954-f011:**
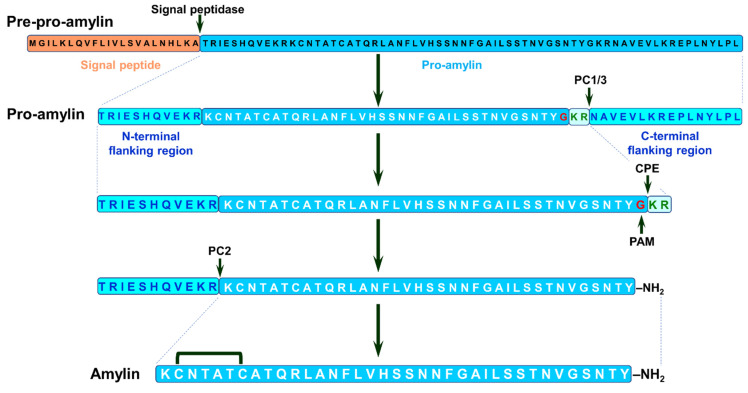
Biosynthesis of human amylin. Amylin is synthetized as 89 a.a. pre-pro-amylin containing a signal peptide (orange box) linked to a pro-amylin precursor (blue box, black residues) containing the final amylin peptide (blue box, white residues), flanked by N- and C-terminal flanking regions (cyan boxes, blue residues). The signal peptide (22 a.a.) is cleaved from pre-pro-amylin in endoplasmic reticulum to produce the pro-amylin peptide (67 a.a.) that is further processed in Golgi and secretory vesicles [102]. Pro-amylin is subsequently cleaved by prohormone convertase 1/3 (PC1/3) to release the C-terminal-flanking region (16 a.a.) and an N-terminally extended pro-amylin intermediate (51 a.a.). Carboxypeptidase E (CPE) cleaves two basic residues (pale blue box, green residues) form C-terminus of 51 a.a. intermediate and subsequently, peptidylglycine alpha-amidating monooxygenase (PAM) oxidizes C-terminal glycine residue (red) with simultaneous amidation of tyrosine carboxyl group. The resulting 48 a.a. intermediate is cleaved by prohormone convertase 2 (PC2) releasing the N-terminal-flanking region (11 a.a.) and mature amylin (37 a.a.). Disulphide bridge is formed between cysteine residues C2 and C7 [102,103]. The cleavage sites for PC1/3, PC2 and CPE, and glycine residue that is the target of PAM are indicated by arrows. Adopted from [102,103].

**Figure 12 life-13-01954-f012:**
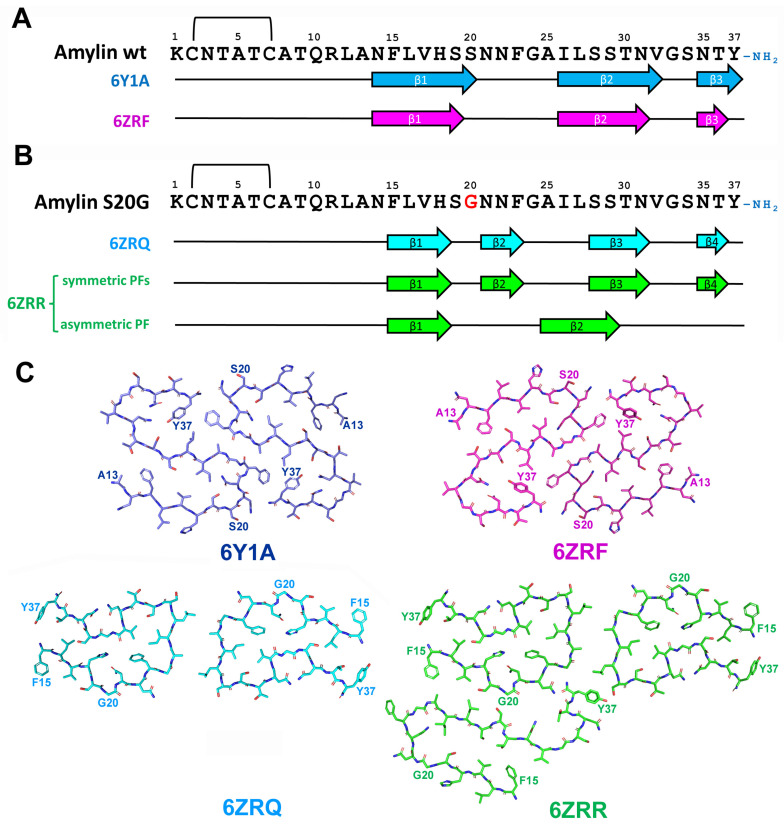
(**A**) Sequence alignment of wild-type amylin with β-strand regions observed in in vitro-aggregated amylin assemblies. The location of β-strands is shown as described in [129] for 6Y1A fold, and in [128] for 6ZRF fold. (**B**) Sequence alignment of wild S20G amylin with β-strand regions observed in in vitro-aggregated amylin assemblies. The location of β-strands is shown as described in [128] for 6ZRQ (**C**). Schematic representation of in vitro-derived wild type and S20G amylin assemblies.

**Figure 13 life-13-01954-f013:**
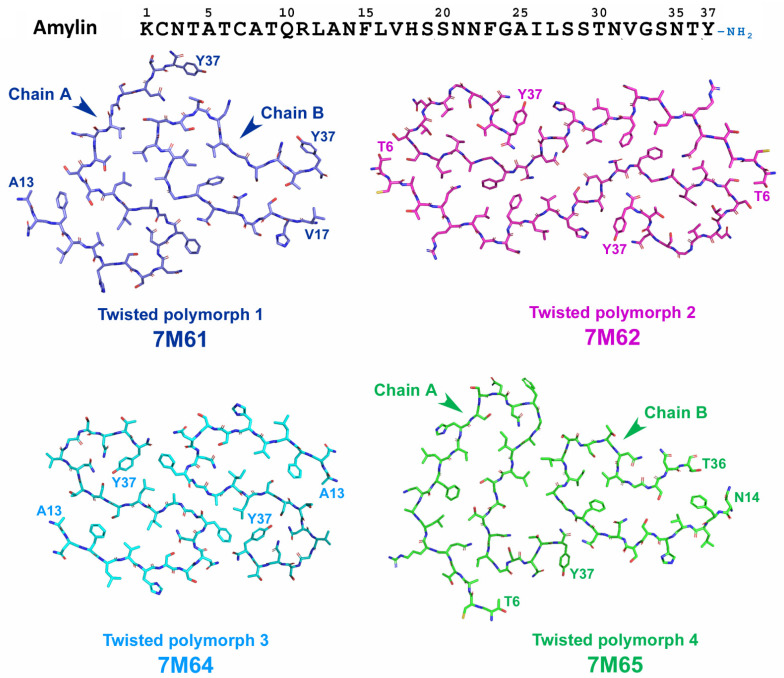
Schematic representation of the structure of wild-type amylin assemblies seeded by T2DM-patient-extracted fibrils [136]. Chain B of Polymorphs 1 and 4 (TW1 and TW4) shows folding patters similar with S20G amylin (symmetric protofibrils of 6ZRR and 6ZRQ folds, shown on Figure 11). Polymorph 3 (TW3) represents typical folding pattern described previously for in vitro assemblies of wild-type amylin (see conformations 6Y1A and 6ZRF on Figure 11). Polymorph 2 (TW2) represents unique folding of amylin that have never been described in previous in vitro experiments and, thus, is the most probable candidate for T2DM-derived conformation of amylin.

**Figure 14 life-13-01954-f014:**
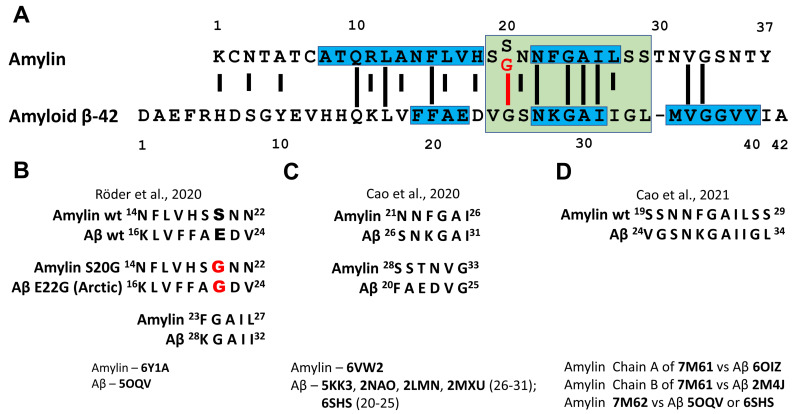
Interaction of amyloid β and amylin. (**A**) Sequence alignment of amylin and amyloid β (adopted from [145]). Long lines indicate identical residues, short lines indicate similarities. S20G mutation in amylin sequence is shown in red and increased identity of amylin and Aβ sequences as a result of S20G substitution is pointed out. Blue boxes display hot regions for Aβ-amylin interaction determined by Andreetto and co-workers [147]. Green box covers the sequences of undecapeptidic fragments of Aβ and S20G-amylin with the highest structural similarity [148]. (**B**–**D**) Structurally similar regions of Aβ and amylin fibrillar assemblies revealed by in silico superimposition of described Aβ and amylin structures stored in RCSB PDB [151]. The RCSB PDB reference code of the structures used in the experiment is mentioned. In model B, described by Röder and co-workers [129], the regions Aβ (16–24) and amylin (14–22) as well as Aβ (28–32) and amylin (23–27) possess similar structure. S20G substitution in amylin and E22G substitution in Aβ should result in analogous conformational consequences in the structure of both amylin and Aβ protofibrils. Model C, described by Cao and co-workers [130] recognize two regions of structural similarity between amylin and Aβ assemblies. In model D, developed by Cao and co-authors [136] the regions of Aβ and amylin responsible for cross-seeding were predicted.

**Table 1 life-13-01954-t001:** Interaction of proteins involved in human amyloidoses.

	Interacting Proteins	Co-Deposition	Cross-Seeding		Co-Aggregation		Interaction of Protein Monomers	
			Fibrillar Seeds	Oligomeric Seeds	Mixed Fibrillar Assemblies	Chimeric Oligomer Assemblies	PTM	Templating
1.	Aβ–amylin	+	+		+			
2.	Aβ–tau						+	+
3.	Aβ–TDP-43			+		+		
4.	Aβ–α-Syn	+	+	+		+		+
5.	Tau–α-Syn	+	+		+	+	+	
6.	Tau–TDP-43			+				
7.	Tau–amylin	+						
8.	α-Syn–TPD-43					+		
9.	α-Syn–amylin	+	+		+	+		+
10.	TDP-43–amylin							

## Data Availability

Not applicable.

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
