# Peer review of "Interaction of Proteins Involved in Neuronal Proteinopathies"

_life, 2023, doi:10.3390/life13101954_

Round 1

Reviewer 1 Report

This manuscript entitled Interaction of proteins involved in neuronal proteinopathies review article address the main pathologies, related to the aggregation of proteins extracted from biological samples and its influence on the progression of underlying disease and comorbidities. I think this manuscript is interesting and comprehensive. It is well-written and organized However I have few concerns regarding it. Please find detailed comments below.

1. First of all it is very long review article about 32 pages. It would be better if the author divides the topics into 2 or 3 manuscripts focusing on specific issue rather than including all of these topics together, it is just confusing.

2. I couldn’t find the clinical implications of amyloid protein aggregation and interaction with other protein, is this can be assessed as a risk factor for disease incidence or progression, can it be applied for therapeutic management or screening  for those neurological diseases?

3. Finally, I have concerns regarding the objective of the manuscript; I think it is not consistent with the main objective of the Journal. It is more focused on the biological disorders associated with neurological diseases.

Reviewer 2 Report

The authors provide a comprehensive overview of aggregated proteins in neurodegenerative diseases. Their explanations of individual protein characteristics and interactions are highly valuable. I suggest the following improvements:

1. The description of Amylin (Section 2.5) is currently limited to its association with type 2 diabetes, lacking clarity on its relevance to neurodegenerative diseases. It would be preferable to focus on the connection between Amylin and neurodegenerative disorders, including the acknowledgment of limited available knowledge. To maintain the overall focus, consider reducing the emphasis on Amylin's role in diabetes.

2. Figures appear to have low resolution. In particular, the fonts within the Figure seem blurry. Please make the necessary corrections.

1. Please insert a comma before "and" when listing more than 3 items. For example, "amyloid β peptide, tau protein, α-synuclein, TDP-43 and amylin" should be "amyloid 19 β peptide, tau protein, α-synuclein, TDP-43, and amylin."

2. Please correct typographical errors, such as the unnecessary period after "α-" in α-synuclein and the misspelling of "β strand" as "β stand."

Round 2

Reviewer 1 Report

Thanks for the comprehensive response. The authors adequately addressed all the raised concerns.